# Lagging strand gap suppression connects BRCA-mediated fork protection to nucleosome assembly through PCNA-dependent CAF-1 recycling

Tanay Thakar [1], Ashna Dhoonmoon[1], Joshua Straka[1], Emily M. Schleicher[1], Claudia M. Nicolae [1] & George-Lucian Moldovan [1] ✉

The inability to protect stalled replication forks from nucleolytic degradation drives genome instability and underlies chemosensitivity in BRCA-deficient tumors. An emerging hallmark of BRCA-deficiency is the inability to suppress replication-associated single-stranded DNA (ssDNA) gaps. Here, we report that lagging strand ssDNA gaps interfere with the ASF1-CAF-1 nucleosome assembly pathway, and drive fork degradation in BRCA-deficient cells. We show that CAF-1 function at replication forks is lost in BRCA-deficient cells, due to defects in its recycling during replication stress. This CAF-1 recycling defect is caused by lagging strand gaps which preclude PCNA unloading, causing sequestration of PCNA-CAF-1 complexes on chromatin. Importantly, correcting PCNA unloading defects in BRCA-deficient cells restores CAF-1-dependent fork stability. We further show that the activation of a HIRA-dependent compensatory histone deposition pathway restores fork stability to BRCA-deficient cells. We thus define lagging strand gap suppression and nucleosome assembly as critical enablers of BRCA-mediated fork stability.

The breast cancer susceptibility factors BRCA1 and BRCA2 act as tumor suppressors, by promoting accurate DNA repair through homologous recombination (HR) and protecting against genomic instability, an enabling hallmark of cancer[1,2]. Germline mutations in the BRCA1 and BRCA2 genes drastically increase the lifetime risk of developing breast and ovarian cancer[3–5]. In addition to mediating DNA double-strand break (DSB) repair by HR, the BRCA proteins also play a critical role in maintaining the integrity of DNA replication forks during replication stress[6,7]. A global response to replication stress is replication fork reversal, which involves the regression of replication forks and the annealing of complementary nascent DNA strands[8–10]. Fork reversal necessitates the engagement of the BRCA pathway to stabilize nascent DNA via the formation of RAD51 nucleofilaments. In the absence of an intact BRCA pathway, nascent DNA at reversed forks becomes susceptible to degradation by nucleases, namely MRE11, EXO1, and DNA2[6,7,11,12]. A direct consequence of fork degradation is the accumulation of DNA damage and gross chromosomal aberrations, making fork protection a major mechanism by which the BRCA pathway maintains genome stability and tumor suppression[6,7]. Importantly, fork degradation is also linked to sensitivity to chemotherapeutic agents, and restoration of fork stability is associated with chemoresistance in BRCA deficient cancers[12].

The sliding DNA clamp proliferating cell nuclear antigen (PCNA) is a core component of the DNA replication machinery. During DNA synthesis, PCNA interacts with DNA polymerases to maintain their engagement on template DNA, thereby increasing their processivity. In addition to polymerase recruitment, PCNA also serves as a scaffold for the recruitment of numerous other replication factors and acts as a

[1]Department of Biochemistry and Molecular Biology, The Pennsylvania State University College of Medicine, Hershey, PA 17033, USA.
✉e-mail: glm29@psu.edu

functional toolbelt for DNA replication[13]. PCNA also orchestrates replication coupled nucleosome assembly by recruiting the histone chaperone chromatin assembly factor-1 (CAF-1)[14,15]. PCNA performs distinct functions during leading and lagging strand DNA replication: On the leading strand, PCNA supports continuous DNA synthesis by recruiting and facilitating the function of Polε. On the lagging strand, PCNA recruits Polδ to synthesize Okazaki fragments (OFs) by extending RNA primers assembled by Polα. Subsequently, PCNA recruits the flap endonuclease FEN1 to cleave downstream RNA primers displaced by Polδ, and the DNA ligase LIG1 to seal the resulting nick to yield intact stretches of DNA[16].

Precise DNA replication requires a tight regulation of PCNA cycling at replication forks. PCNA is loaded during replication initiation by the RFC1-5 complex and unloaded by an RFC-like complex composed of ATAD5 and RFC2-5 upon replication completion[17–19]. On lagging strands, frequent Polα mediated repriming necessitates the constant loading of PCNA homotrimers to support the synthesis of multiple OFs. PCNA is unloaded by ATAD5 from the lagging strand upon OF maturation. OF ligation by LIG1 is an essential prerequisite for PCNA unloading, and a failure to unload PCNA can drive genome instability by sequestering PCNA interacting factors at inactive replication factories[18,20,21].

RAD18-mediated ubiquitination of PCNA at the lysine 164 (K164) residue is a prominent response of eukaryotic cells to replication stress. This modification enables the post replicative repair (PRR) of ssDNA gaps through translesion synthesis (TLS)[22–26]. By generating PCNA-K164R mutant human cell lines, completely deficient in PCNA ubiquitination, we recently uncovered an essential role of ubiquitinated PCNA in preventing the nucleolytic degradation of stalled replication forks[27]. Mechanistically, we showed that fork degradation in PCNA-K164R cells is caused by the accumulation of lagging strand gaps, which sequester PCNA as OF ligation is impaired. Since CAF-1 forms a tight complex with PCNA, it is also sequestered in these PCNA complexes in the wake of replication forks, thus impeding replication-coupled nucleosome assembly in these cells, and priming stressed forks for nucleolytic degradation.

Recent publications have revealed a previously underappreciated role of the BRCA-RAD51 pathway in suppressing the accumulation of replication-associated single-stranded DNA (ssDNA) gaps. Gap mitigation by the BRCA pathway occurs through two distinct mechanisms: (1) by restraining fork progression during replication stress, thereby suppressing excessive PRIMPOL-mediated fork repriming[28–32]; and (2) by promoting RAD51-dependent PRR of gaps[27,33–36]. Importantly, replication-associated gaps have been connected to PARP inhibitor (PARPi) sensitivity in BRCA-deficient cells[27,30,31,37,38]. Interestingly, Okazaki fragment processing defects have also been identified in BRCA-deficient cells[37,38].

Here, we show that nucleosome assembly controls fork protection, genomic stability, and chemoresistance of BRCA-deficient cells. Mechanistically, we show that Polα-dependent lagging strand ssDNA gaps cause CAF-1 recycling defects in BRCA-deficient cells, since they sequester PCNA-CAF-1 complexes behind replication forks. The subsequent reduction in CAF-1 availability at ongoing replication forks underlies fork degradation in BRCA-deficient cells. Indeed, we demonstrate that correcting PCNA unloading defects restores fork protection to BRCA-deficient cells in a CAF-1-dependent manner, thereby exposing efficient PCNA unloading and CAF-1-mediated nucleosome assembly as major effectors of fork protection by the BRCA-RAD51 pathway. Moreover, we show that loss of CAF-1 restores fork protection to BRCA-deficient cells by releasing an alternative nucleosome assembly pathway mediated by the histone chaperone HIRA. Our work uncovers an unexpected role for nucleosome deposition pathways in mediating BRCA-dependent genome stability.

## Results

### CAF-1 loss rescues fork stability in BRCA-deficient cells

We recently showed that replication-coupled nucleosome assembly mediated by CAF-1 is critical for the stability of stalled replication forks[27]. We therefore sought to investigate the effect of CAF-1 inactivation on fork stability in cells deficient in either BRCA1 or BRCA2 function. To assess fork stability, we subjected cells to consecutive labeling with the nucleotide analogs IdU and CldU, respectively, for 30 min each followed by fork arrest with 4 mM hydroxyurea (HU) for 4 h. Fork stability was investigated by measuring the ratios of CldU tract lengths to adjacent IdU tract lengths, allowing us to control for potential changes in fork speed brought about by CAF-1 inactivation. Strikingly, depletion of CHAF1A (the largest subunit of the CAF-1 complex, also known as p150) fully restored fork stability to HeLa-BRCA2KO as well as to RPE1-p53KOBRCA1KO cells, while causing fork degradation in their respective BRCA-proficient counterparts (Fig. 1a, b; Supplementary Fig. 1a, b). To rule out potential siRNA off-target effects, we employed CRISPR/Cas9 to knock-out CHAF1A in 293T and HeLa cells (Supplementary Fig. 1c, d). Similar to CHAF1A depletion using siRNA, both 293T and HeLa CHAF1A-knockout cells displayed fork degradation upon HU treatment (Fig. 1c, d). Importantly, BRCA2 depletion caused fork degradation in wild type, but not in CHAF1A-knockout HeLa and 293T cells (Fig. 1c, d; Supplementary Fig. 1e, f). These findings indicate that loss of CAF-1 promotes fork degradation in wild-type cells but suppresses this degradation in BRCA-deficient cells.

Restoration of RAD51 loading on chromatin promotes fork protection in BRCA-deficient settings[39,40]. We therefore sought to assess the impact of CHAF1A inactivation on chromatin-bound RAD51 levels in BRCA1 and BRCA2-depleted cells. Upon treatment with the topoisomerase I inhibitor camptothecin (CPT), depletion of CHAF1A did not affect RAD51 foci formation in BRCA-proficient HeLa cells and failed to ameliorate the reduction in RAD51 foci formation observed in BRCA1 and BRCA2 depleted cells (Fig. 1e, f; Supplementary Fig. 1g). Treatment with the RAD51 inhibitor B02 was previously shown to elicit nascent DNA resection at stalled forks in BRCA-proficient cells[39,41]. In line with this, treatment with B02 resulted in HU-induced fork degradation in wild-type HeLa and 293T cells; in contrast, B02 did not cause fork degradation in HeLa-BRCA2KO cells depleted of CHAF1A (Fig. 1g), or in 293T-CHAF1AKO cells depleted of BRCA2 (Fig. 1h). We previously showed that loss of E2F7 restores fork protection in BRCA2-deficient cells by promoting BRCA2-independent loading of RAD51 on chromatin[40]. In line with this, and in contrast to the observations with CHAF1A, B02 treatment restored fork degradation in E2F7-depleted BRCA2-knockout cells. Collectively, these results indicate that, unlike E2F7 inactivation, CHAF1A inactivation restores fork stability to BRCA-deficient cells in a RAD51-independent manner.

Reversal of stalled replication forks is an essential prerequisite to nascent DNA resection in BRCA-deficient cells[11,34,41,42]. Thus, we investigated if the suppression of fork degradation observed upon CHAF1A depletion in BRCA-deficient cells simply reflects a defect in fork reversal. PARP1 is a critical enabler of fork reversal[43]. The presence of PARP1 at nascent DNA has been previously used as an indirect readout for fork reversal[44]. Under prolonged replication arrest, stable reversed replication forks are marked by PARP1. In contrast, forks undergoing resection lose nascent DNA on regressed arms and no longer retain the structural configuration resembling four-way junctions, thus precluding the presence of PARP1. We therefore used the SIRF (in situ detection of proteins at replication forks) assay, a proximity ligation-based approach (PLA)[41,45], to assess PARP1 binding to nascent DNA upon HU treatment. To account for potential baseline discrepancies arising from variabilities in EdU uptake due to gene knockdowns or drug treatments, we normalized the PARP1-biotin PLA signal in each condition using the fluorescence signal from the corresponding biotin-biotin control (signifying EdU uptake). Similar to inhibition of MRE11 using mirin, depletion of

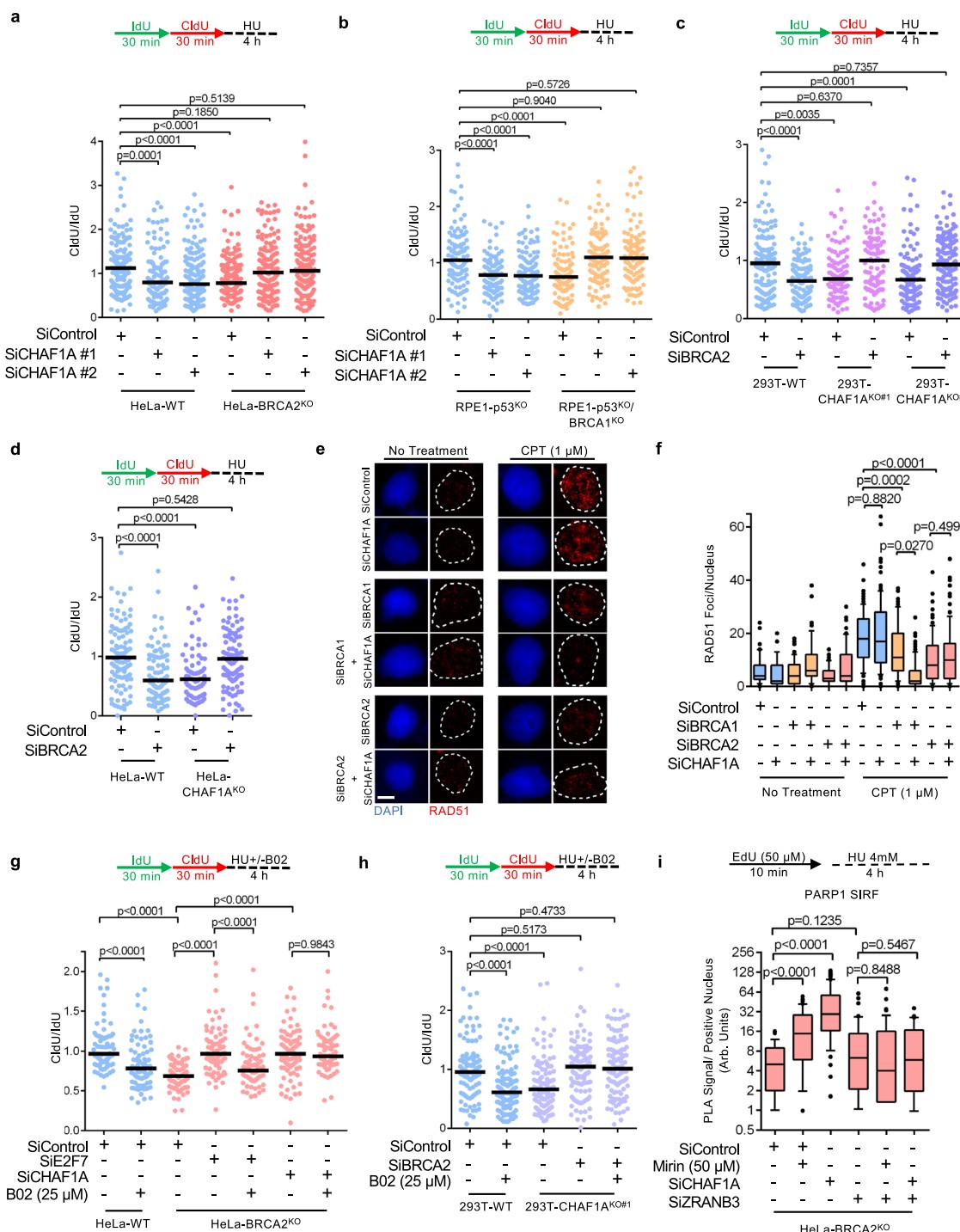

CHAF1A increased PARP1 levels at nascent DNA in HeLa BRCA2[KO] cells (Fig. 1i; Supplementary Fig. 1h, i). Importantly, abolishing fork reversal by depleting the fork remodeling translocase ZRANB3 restored PARP1 levels to similar levels across all conditions, indicating no baseline differences in PARP1 recruitment to nascent DNA (Fig. 1i; Supplementary Fig. 1h, i). These results suggest that loss of CHAF1A does not preclude fork reversal, but rather promotes the stability to reversed forks in BRCA-deficient cells.

## Loss of CAF-1 drives chemoresistance in BRCA-deficient cells

Previous work showed that fork degradation drives DNA damage-induced chromosomal rearrangements in BRCA-deficient cells[6,7,12]. Thus, we next investigated if CHAF1A inactivation could avert DNA damage accumulation in BRCA-deficient cells. We assessed replication-

coupled DNA damage by immunofluorescence detection of γH2AX in cells treated with CPT, known to elicit nascent strand degradation[41]. Indeed, while BRCA1 and BRCA2-depleted HeLa cells exhibited increased γH2AX levels upon CPT treatment, γH2AX levels were ameliorated upon co-depletion of CHAF1A (Fig. 2a, b; Supplementary Fig. 2a). Similar to the observations with HU treatment, co-depletion of CHAF1A in BRCA1 or BRCA2-depleted cells also rescued CPT-induced fork degradation (Supplementary Fig. 2b). We also measured the accumulation of DNA double-stranded breaks (DSBs) in these cells, using the neutral comet assay. Similar to γH2AX induction, treatment with CPT resulted in increased comet tail moments in BRCA1 and BRCA2-depleted HeLa cells, which was rescued upon co-depletion of CHAF1A (Fig. 2c, d). Similar results were obtained upon cisplatin treatment (Fig. 2e). Overall, these findings show that restoration of

**Fig. 1 | Loss of CAF-1 promotes fork stability in BRCA-deficient cells. a, b** DNA fiber combing assays showing that CHAF1A depletion results in HU-induced fork resection in wild-type cells, but suppresses this degradation in BRCA2-knockout HeLa cells (**a**) and in BRCA1-knockout RPE1 cells (**b**). The ratio of CldU to IdU tract lengths is presented, with the median values marked on the graph. The *p*-values (Mann–Whitney test, two-tailed) are listed at the top. Schematic representations of the DNA fiber combing assay conditions are also presented. Western blots confirming the knockdown are shown in Supplementary Fig. 1a, b. **c, d** DNA fiber combing assays showing that CHAF1A knockout in 293T (**c**) or HeLa (**d**) cells results in HU-induced fork degradation, which is suppressed by BRCA2 knockdown. The ratio of CldU to IdU tract lengths is presented, with the median values marked on the graph. The *p*-values (Mann–Whitney test, two-tailed) are listed at the top. Schematic representations of the DNA fiber combing assay conditions are also presented. Western blots confirming the knockdown are shown in Supplementary Fig. 1c–f. **e, f** RAD51 immunofluorescence experiment showing that CHAF1A depletion does not restore CPT-induced RAD51 foci in BRCA1 or BRCA2-depleted cells. HeLa cells were treated with 1 μM CPT for 1 h followed by media removal and chase in fresh media for 3 h. Representative micrographs (**e**) and quantifications (**f**) are shown (scale bar represents 10 μm). At least 50 cells were quantified for each

condition. Center line indicates the median, bounds of box indicate the first and third quartile, and whiskers indicate the 10th and 90th percentile. The *p*-values (Mann–Whitney test, two-tailed) are listed at the top. Western blots confirming the co-depletions are shown in Supplementary Fig. 1g. **g, h** Inhibition of RAD51 by B02 treatment does not restore HU-induced fork degradation in CHAF1A-depleted HeLa-BRCA2^KO cells (**g**), or in BRCA2-depleted 293T-CHAF1A^KO cells (**h**). In contrast, B02 treatment restores HU-induced fork degradation in E2F7-depleted HeLa-BRCA2^KO cells (**g**). The ratio of CldU to IdU tract lengths is presented, with the median values marked on the graph. The *p*-values (Mann–Whitney test, two-tailed) are listed at the top. Schematic representations of the DNA fiber combing assay conditions are also presented. **i** SIRF assay showing that PARP1 binding to nascent DNA is increased upon CHAF1A depletion in HeLa-BRCA2^KO cells, indicating stabilization of reversed replication forks. At least 40 positive cells were quantified for each condition. Center line indicates the median, bounds of box indicate the first and third quartile, and whiskers indicate the 10th and 90th percentile. The *p*-values (Mann–Whitney test, two-tailed) are listed at the top. A schematic representation of the SIRF assay conditions is also presented. Representative micrographs (scale bar represents 10 μm) are shown in Supplementary Fig. 1h. Western blots confirming the co-depletions are shown in Supplementary Fig. 1i.

fork stability to BRCA-deficient cells upon CHAF1A depletion is associated with suppression of DNA damage acumulation in these cells.

Restoration of fork stability is considered a driver of chemoresistance in BRCA-deficient cells[12,46]. By employing clonogenic survival assays, we next investigated the impact of CHAF1A inactivation in BRCA-deficient cells on cisplatin sensitivity. CHAF1A co-depletion significantly rescued cisplatin sensitivity in Hela cells depleted of BRCA1 or BRCA2 (Fig. 2f, g). Cisplatin chemotherapy is the mainstay therapeutic approach in ovarian cancer treatment. We thus investigated if CHAF1A levels impact the chemosensitivity of BRCA-mutant ovarian tumors in clinical samples. Analyses of survival and matched genotype and expression data from TCGA datasets indicated that CHAF1A expression can stratify the survival of individuals with BRCA2-mutant ovarian tumors: high CHAF1A expression trended towards increased survival, while low CHAF1A expression trended towards reduced survival (Supplementary Fig. 2c). This is in line with our clonogenic survival results showing that CHAF1A depletion causes cisplatin resistance in BRCA2-deficient cells. Taken together, these observations suggest that CHAF1A inactivation can enable BRCA-deficient cells to avert replication-coupled DNA damage, thereby driving chemoresistance and potentially exacerbating adverse clinical outcomes in patients with BRCA1/2 mutated cancers.

### Nucleosome assembly ensures replication fork protection

Since the cellular function of CAF-1 is in nucleosome deposition, we next sought to investigate if nucleosome assembly is a general determinant of fork stability in BRCA-deficient cells. The histone chaperone anti-silencing factor 1 (ASF1) operates upstream of two distinct nucleosome assembly mechanisms: a CAF-1-dependent co-replicational process depositing the H3 isoform H3.1, and replication-independent processes involving the histone chaperones HIRA and DAXX, depositing the H3.3 isoform[47–50]. To test if inactivating ASF1 rescues fork stability in BRCA-deficient cells, we depleted ASF1A (one of the two human ASF1 paralogs) either alone, or in conjunction with BRCA1 or BRCA2 in HeLa cells. ASF1A knockdown elicited nascent DNA resection in BRCA-proficient cells, but, in contrast to CHAF1A depletion, failed to rescue fork resection in cells depleted either of BRCA1 or BRCA2 (Fig. 3a; Supplementary Fig. 3a). This suggests that alternative ASF1A-dependent nucleosome assembly pathways could compensate for CHAF1A inactivation in BRCA-deficient cells, to restore fork stability. Indeed, co-depleting CHAF1A and ASF1A in HeLa BRCA2^KO cells restored fork degradation, suggesting that fork stability upon CHAF1A inactivation in BRCA-deficient cells depends on ASF1 (Fig. 3b; Supplementary Fig. 3b–d). These results suggest that ASF1-

dependent nucleosome assembly is an essential component of fork protection and determines fork stability in the context of BRCA deficiency.

The observed epistasis between the inactivation of ASF1A and BRCA proteins led us to examine if ASF1A-dependent nucleosome assembly elicits fork protection through mechanisms similar to the BRCA pathway. Loss of BRCA1 or BRCA2 function renders stalled forks susceptible to resection by the MRE11 and DNA2 nucleases[6,7,12,29]. Similar to this, depletion of either CHAF1A or ASF1A in HeLa cells elicited nascent DNA resection, which could be rescued by inhibition of MRE11 or DNA2 using the small molecule inhibitors mirin and C5 respectively (Fig. 3c). Next, we tested if fork reversal was also required for fork degradation in this context. Fork degradation in CHAF1A-knockdown HeLa cells, as well as in CHAF1A-knockout 293T cells, was rescued upon depletion of the fork remodeling enzymes SMARCAL1 and ZRANB3 (Fig. 3c, d; Supplementary Fig. 3e, f). These results indicate that nucleosome assembly is a general determinant of replication fork stability, and that the inactivation of nucleosome assembly elicits fork resection through mechanisms similar to those operating in BRCA-deficient cells.

### Fork stability restoration requires compensatory nucleosome assembly

ASF1A was shown to be part of two mutually exclusive nucleosome assembly complexes involving the CAF-1 and HIRA histone chaperones, responsible for the deposition of H3.1 and H3.3 respectively[50,51]. HIRA-mediated histone deposition was previously shown to compensate for the inactivation of CAF-1-mediated replication-dependent nucleosome assembly[52]. Unlike ASF1A, we found that ASF1B does not impact fork degradation in wild-type or BRCA-deficient cells (Supplementary Fig. 3c, d). Previous studies have shown that ASF1A, but not ASF1B, interacts with HIRA[50]. Because our findings suggest a HIRA-dependent rescue of fork protection, and since restoration of replication fork stability is an important component of cell survival in BRCA-deficient cells[46,53], we assessed if HIRA promotes cell survival in BRCA-deficient cells upon CHAF1A inactivation. We queried publicly available CRISPR screening data for the relative dependence of BRCA1-deficient cells on HIRA and CHAF1A. We observed a linear regression of CHAF1A and HIRA gene dependency scores (CERES; lower scores correspond to higher dependencies), showing that BRCA1-proficient cells tend to be dependent on both CHAF1A and HIRA for survival (Fig. 4a). This implies a general pattern of reliance on nucleosome assembly pathways. Strikingly, in cells carrying deleterious BRCA1 mutations, a lower survival dependency on CHAF1A correlated with a greater dependency on HIRA and vice-versa, suggesting that

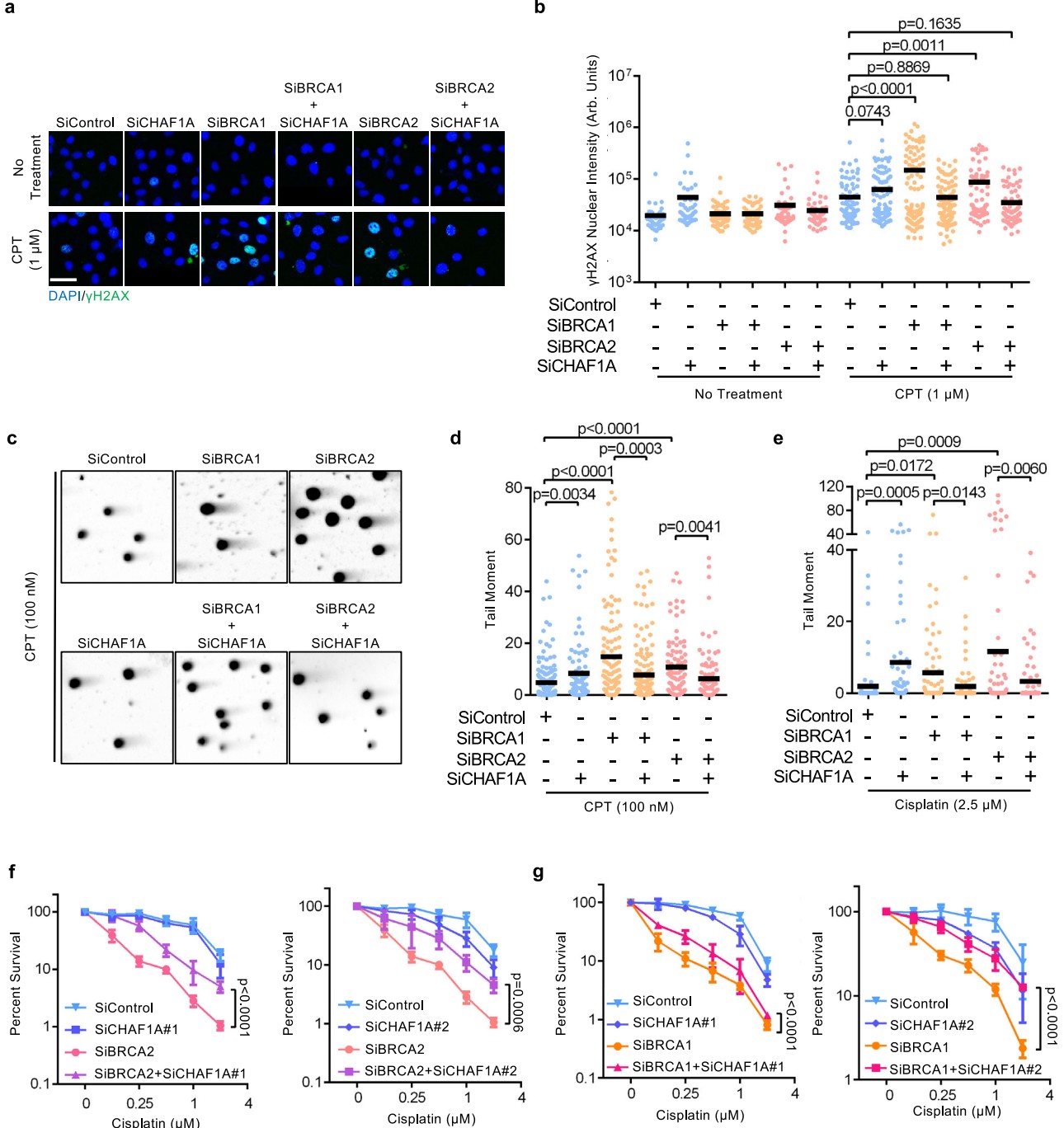

**Fig. 2 | Loss of CAF-1 suppresses genomic instability in BRCA-deficient cells.**
**a**, **b** γH2AX immunofluorescence experiment showing that CHAF1A depletion suppresses CPT-induced DNA damage accumulation in BRCA1 or BRCA2-depleted cells. HeLa cells were treated with 1 μM CPT for 1 h followed by media removal and chase in fresh media for 3 h. Representative micrographs (scale bar represents 50 μm) (**a**) and quantifications (**b**) are shown. At least 50 cells were quantified for each condition. The mean values are represented on the graph, and the *p*-values (*t*-test, two-tailed, unpaired) are listed at the top. Western blots confirming the co-depletions are shown in Supplementary Fig. 2a. **c**–**e** Neutral comet assay showing that CHAF1A depletion suppresses CPT-induced (**c**, **d**) and cisplatin-induced (**e**)

DSB formation in BRCA1 or BRCA2-depleted cells. HeLa cells were treated with 100 nM CPT for 4 h or 2.5 μM cisplatin for 24 h. Representative micrographs (**c**) and quantifications (**d**, **e**) are shown. At least 60 nuclei were quantified for each condition. The mean values are represented on the graph, and the *p*-values (*t*-test, two-tailed, unpaired) are listed at the top. **f**, **g** Clonogenic survival experiments showing that CHAF1A co-depletion in BRCA2-knockdown (**f**) or BRCA1-knockdown (**g**) HeLa cells promotes cisplatin resistance. The average of three experiments, with standard deviations indicated as error bars, is shown. Asterisks indicate statistical significance (two-way ANOVA).

BRCA1-deficient cells rely on HIRA for cell survival in the absence of CHAF1A (Fig. 4a). It was previously shown that the histone chaperone DAXX can also cooperate with ASF1 in H3.3-dependent nucleosome assembly[47]. In contrast to HIRA, an increased dependence on CHAF1A did not correlate with increased DAXX dependency in BRCA1-proficient cells (Fig. 4b). Moreover, in BRCA1-deficient cells, an increased dependency on DAXX was not associated with a reduced dependency on CHAF1A (Fig. 4b). These observations suggest that upon CHAF1A inactivation, BRCA-deficient cells rely on HIRA and not on DAXX to mediate nucleosome assembly and ensure cell survival.

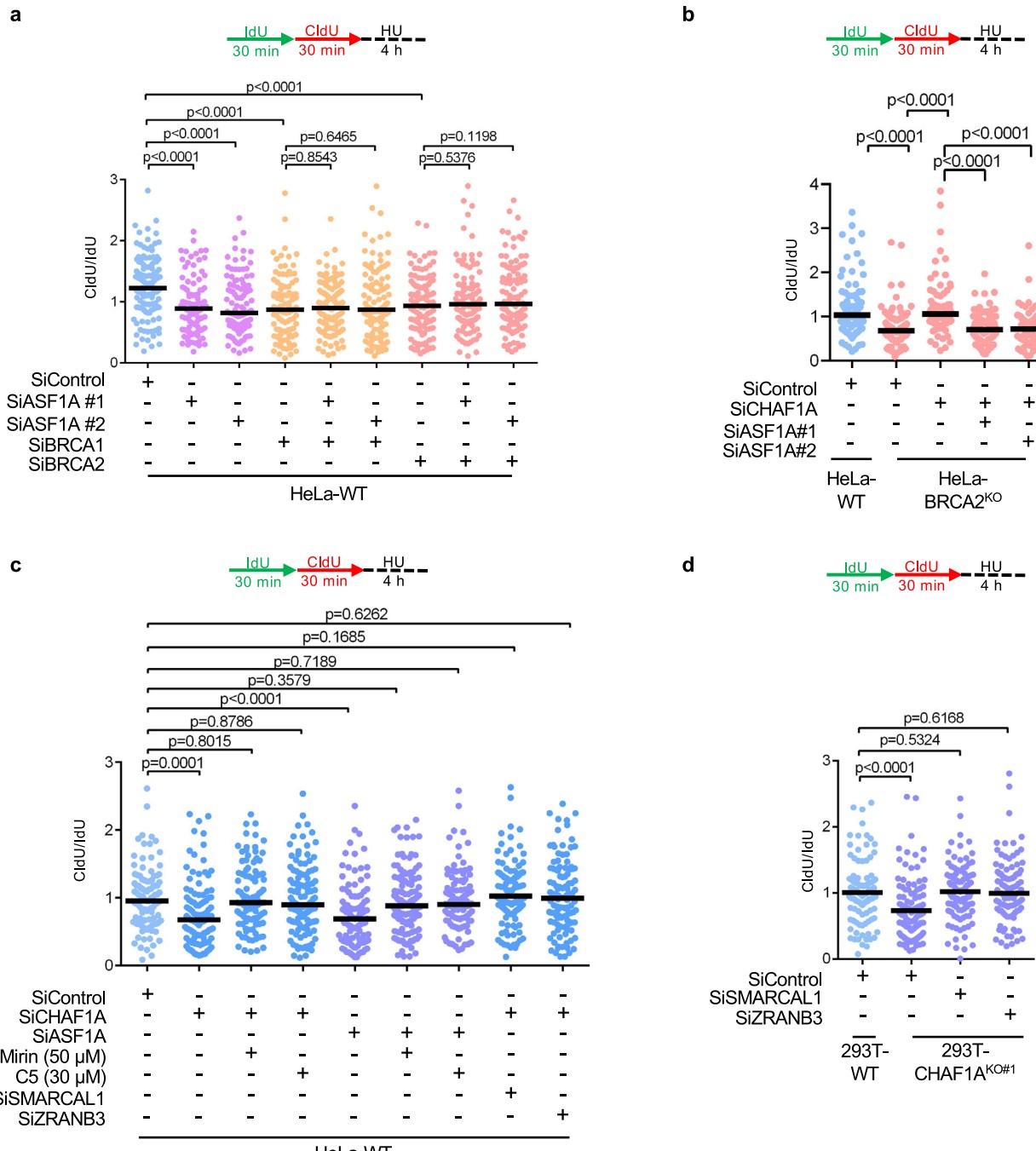

**Fig. 3 | Determinants of CHAF1A-mediated fork protection. a** DNA fiber combing assay showing that ASF1A depletion results in HU-induced fork degradation in wild-type cells, but does not affect this degradation in BRCA1 or BRCA2-depleted HeLa cells. The ratio of CldU to IdU tract lengths is presented, with the median values marked on the graph. The *p*-values (Mann–Whitney test, two-tailed) are listed at the top. A schematic representation of the DNA fiber combing assay conditions is also presented. Western blots confirming the co-depletion are shown in Supplementary Fig. 3a. **b** DNA fiber combing assay showing that ASF1A co-depletion restores HU-induced fork degradation in CHAF1A-knockdown HeLa-BRCA2^KO cells. The ratio of CldU to IdU tract lengths is presented, with the median values marked on the graph. The *p*-values (Mann–Whitney test, two-tailed) are listed at the top. A schematic representation of the DNA fiber combing assay conditions is also presented. Western blots confirming the co-depletion are shown in Supplementary Fig. 3b. **c** DNA

fiber combing assay showing that inhibition of nucleases MRE11 (by treatment with mirin) or DNA2 (by treatment with C5), or co-depletion of DNA translocases SMARCAL1 and ZRANB3, suppresses HU-induced fork degradation caused by CHAF1A or ASF1A loss in HeLa cells. The ratio of CldU to IdU tract lengths is presented, with the median values marked on the graph. The *p*-values (Mann–Whitney test, two-tailed) are listed at the top. A schematic representation of the DNA fiber combing assay conditions is also presented. Western blots confirming the co-depletion are shown in Supplementary Fig. 3e. **d** DNA fiber combing assay showing that SMARCAL1 or ZRANB3 depletion suppresses HU-induced fork degradation in 293T-CHAF1A^KO cells. The *p*-values (Mann–Whitney test, two-tailed) are listed at the top. A schematic representation of the DNA fiber combing assay conditions is also presented. Western blots confirming the knockdowns are shown in Supplementary Fig. 3f.

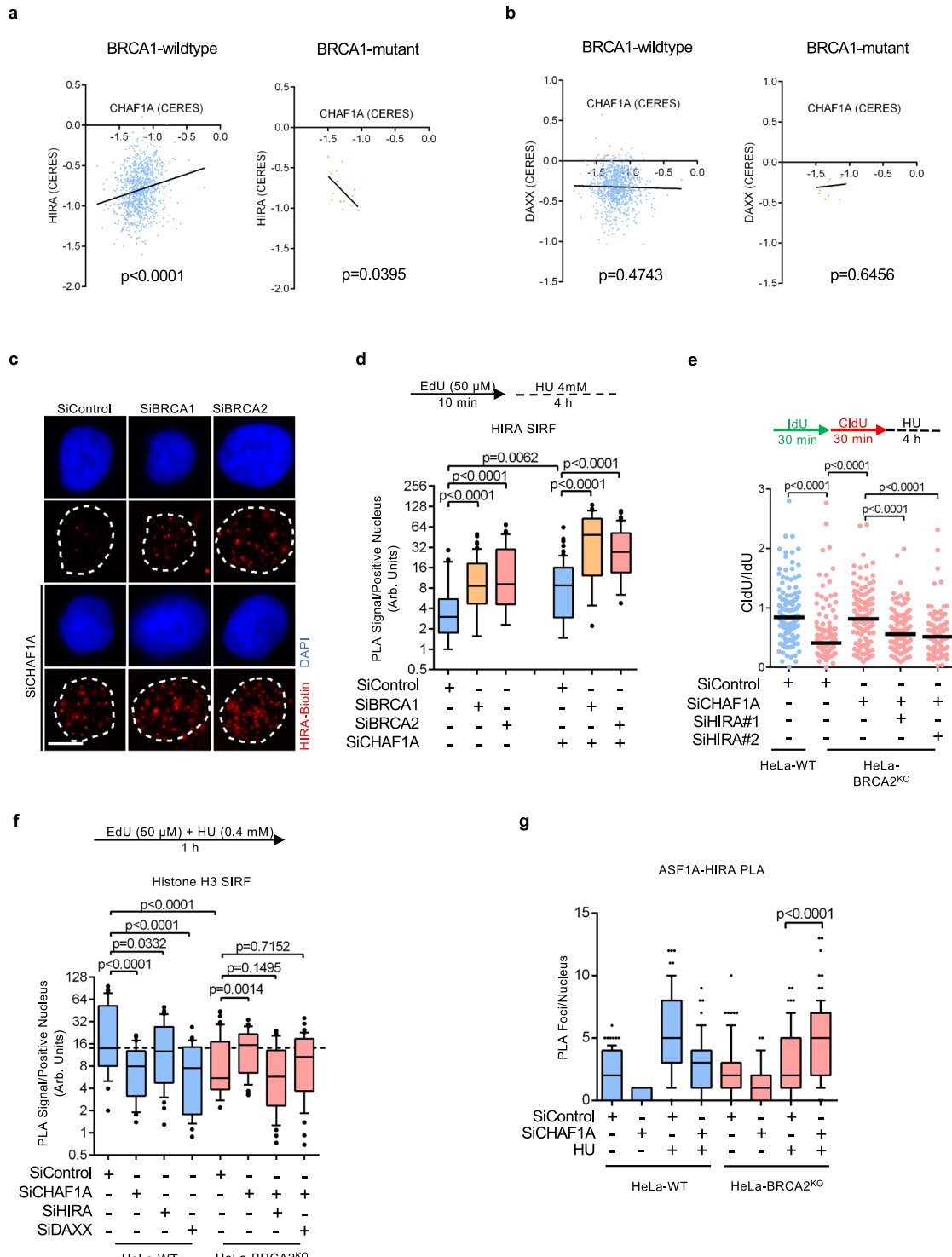

We next investigated if the recruitment of HIRA to stalled forks was differentially regulated in BRCA-deficient cells. SIRF assays showed that depletion of BRCA1 or BRCA2 results in increased recruitment of HIRA to HU-stalled forks, but this was not the case in the absence of HU treatment (Fig. 4c, d; Supplementary Fig. 4a, b). As a control, siRNA-mediated depletion of HIRA in HeLa cells resulted in an acute decrease in SIRF signal, confirming the specificity of this approach in detecting HIRA binding to nascent DNA (Supplementary Fig. 4a). Importantly, while CHAF1A depletion resulted in an overall increase in HIRA levels at HU-stalled forks, BRCA-deficient cells still displayed an enhanced recruitment of HIRA compared to BRCA-proficient cells (Fig. 4c, d), suggesting that BRCA1/2 inactivation guides the differential presence

of HIRA at stalled replication forks. In contrast to HIRA, DAXX levels at stalled replication forks in BRCA1/2-depleted cells showed no change compared to BRCA-proficient cells (Supplementary Fig. 4c), suggesting that HIRA rather than DAXX may be preferentially operational at stalled forks in BRCA-deficient cells.

Based on these findings, we next investigated if the HIRA-H3.3 pathway was responsible for restoring nucleosome assembly and fork stability to BRCA-deficient cells upon CHAF1A loss. Indeed, co-depletion of HIRA or of H3.3 reversed the fork rescue elicited by CHAF1A knockdown in BRCA2$^{KO}$ cells (Fig. 4e; Supplementary Fig. 4d–f). Next, we employed the SIRF assay to measure histone H3 loading on nascent DNA, using an antibody that recognizes both H3.1

**Fig. 4 | Fork protection upon CAF-1 loss in BRCA-deficient cells requires the histone chaperone HIRA. a, b** Linear regressions of (**a**) HIRA Gene Effect (CERES) vs. CHAF1A Gene Effect (CERES) and (**b**) DAXX Gene Effect vs. CHAF1A Gene Effect (CERES) in Cancer Cell Line Encyclopedia (CCLE) cell lines containing either wild-type BRCA1 or BRCA1 with deleterious mutations are shown. A lower CERES score corresponds to greater survival dependency. **c, d** SIRF assay showing that HIRA binding to nascent DNA is increased upon BRCA1 or BRCA2 depletion in HeLa cells. Representative micrographs (scale bar represents 10 μm) (**c**) and quantifications (**d**) are shown. At least 30 positive cells were quantified for each condition. Center line indicates the median, bounds of box indicate the first and third quartile, and whiskers indicate the 10th and 90th percentile. The *p*-values (Mann–Whitney test, two-tailed) are listed at the top. A schematic representation of the SIRF assay conditions is also presented. **e** DNA fiber combing assay showing that HIRA co-depletion restores HU-induced nascent strand resection in CHAF1A-knockdown HeLa-BRCA2^KO cells. The ratio of CldU to IdU tract lengths is presented, with the median values marked on the graph. The *p*-values (Mann–Whitney test, two-tailed)

are listed at the top. A schematic representation of the DNA fiber combing assay conditions is also presented. Western blots confirming the co-depletion are shown in Supplementary Fig. 4d. **f** SIRF assay showing that histone H3 binding to nascent DNA is reduced in BRCA2-knockout cells, but restored in a HIRA-dependent manner upon CHAF1A depletion. At least 35 positive cells were quantified for each condition. Center line indicates the median, bounds of box indicate the first and third quartile, and whiskers indicate the 10th and 90th percentile. The *p*-values (Mann–Whitney test, two-tailed) are listed at the top. A schematic representation of the SIRF assay conditions is also presented. **g** PLA assay showing that the interaction between ASF1A and HIRA is enhanced upon concomitant BRCA2 and CHAF1A inactivation in HeLa cells treated with HU (4 mM for 3 h). At least 75 cells were quantified for each condition. Center line indicates the median, bounds of box indicate the first and third quartile, and whiskers indicate the 10th and 90th percentile. The *p*-values (Mann–Whitney test, two-tailed) are listed at the top. Experiments with knockdown of ASF1A or HIRA, demonstrating the specificity of the signal, are shown in Supplementary Fig. 4i.

and H3.3 isoforms. We found that BRCA2-knockout cells have reduced histone H3 incorporation on nascent DNA upon concurrent HU treatment, similar to the reduction induced by CHAF1A depletion in wild-type cells, thus confirming that BRCA-deficient cells are defective in nucleosome assembly upon experiencing replication stress (Fig. 4f). CHAF1A depletion in BRCA-knockout cells restored histone H3 levels on nascent DNA, and this was dependent on HIRA (Fig. 4f), arguing that HIRA is responsible for restoring nucleosome assembly and fork protection in BRCA-deficient cells upon loss of CAF-1. HIRA depletion in wild-type cells had only a minimal effect on H3 incorporation during replication stress (Fig. 4f), suggesting that its role in nucleosome assembly at stressed forks is only activated upon concurrent BRCA and CAF-1 inactivation.

To further assess the fork-protective properties of HIRA, we depleted HIRA in wild-type HeLa cells. Unlike the depletion of ASF1A, CHAF1A, or DAXX, HIRA knockdown did not elicit nascent DNA resection upon fork stalling (Supplementary Figs. 3c, S4g, h). This was in line with our SIRF experiments (Fig. 4f), suggesting that HIRA-mediated fork protection is selectively activated during CHAF1A loss in BRCA-deficient cells. To address this, we employed proximity ligation assays to investigate the interaction between ASF1A and HIRA. We found that, in BRCA2-deficient cells, CHAF1A depletion enhances ASF1A co-localization with HIRA upon HU treatment (Fig. 4g; Supplementary Fig. 4i). Altogether, these findings suggest that loss of CAF-1 triggers HIRA-mediated nucleosome assembly to protect stalled replication forks in BRCA-deficient cells.

## BRCA-deficient cells exhibit CAF-1 recycling defects

Since HIRA-mediated fork protection in BRCA1/2-deficient cells is triggered only upon the inactivation of CHAF1A, we sought to track the dynamics of CHAF1A at stalled forks in these cells. Replication fork stalling is accompanied by the unloading of PCNA and CAF-1[54,55]. SIRF experiments showed similar CHAF1A levels at unperturbed replication forks in BRCA-proficient and BRCA1/2-depleted HeLa cells (Fig. 5a, b; Supplementary Fig. 5a). However, upon HU-induced replication arrest, BRCA-proficient cells showed lower levels of CHAF1A on EdU-labeled DNA, while CHAF1A levels remained virtually unchanged in BRCA1/2-depleted cells (Fig. 5a, b), suggesting a potential defect in CAF-1 unloading from stalled forks in BRCA-deficient cells.

We recently showed that in PCNA ubiquitination-deficient K164R cells, the chromatin unloading of PCNA-CAF-1 complexes is defective since these complexes are retained at spontaneously accumulating ssDNA gaps on the lagging strand, which preclude Okazaki fragment maturation behind replication forks[27]. Interestingly, recent work from several laboratories demonstrated that BRCA-deficient cells are also prone to gap accumulation during replication stress, owing to their inability to restrain replication fork progression[28–30,37]. Altogether, these findings led us to hypothesize that replication stress-induced

ssDNA gap accumulation in BRCA-deficient cells may retain CAF-1 on lagging strands, similar to the situation we previously described in PCNA-K164R cells. To test this, we performed SIRF experiments with EdU labeling in the presence of a low dose of HU which elicits gap formation but does not result in replication fork arrest[30]. While the recruitment of CHAF1A to replication forks remained unchanged, BRCA1/2-depleted cells showed a persistent retention of CHAF1A at EdU-labeled DNA after a 4 h thymidine chase (Fig. 5c). Importantly, CHAF1A retention defects were not observed in BRCA1/2-depleted cells under endogenous (non-HU treatment) conditions (Supplementary Fig. 5a). In contrast, the in situ inactivation of RAD51 function using simultaneous treatments with B02 and HU, as opposed to the prior genetic inactivation of BRCA1/2, was enough to elicit CHAF1A unloading defects in BRCA-proficient HeLa cells (Fig. 5d). In addition, HU-induced fork degradation caused by B02 treatment could also be rescued by CHAF1A depletion (Fig. 5e), similar to what we observed in BRCA-deficient cells. These findings argue that the inability of BRCA-deficient cells to suppress ssDNA gaps during replication stress drives abnormal CHAF1A retention at stalled replication forks.

We recently showed that PCNA-dependent sequestration of CHAF1A at gaps behind replication forks drives nucleosome assembly defects and fork degradation in PCNA ubiquitination-deficient K164R cells[27]. We thus hypothesized that, in BRCA-deficient cells, CAF-1 chromatin retention at replication stress-induced ssDNA gaps left behind forks, reduces its availability at ongoing replication forks; this would cause nucleosome deposition defects, and prime nascent DNA for degradation upon fork stalling. To test this, we pre-treated cells with a low dose of HU, and investigated the recruitment of CHAF1A at EdU-labeled DNA after a 4 h chase. CHAF1A SIRF signal was significantly lower in BRCA1/2-depleted cells than in BRCA-proficient cells (Fig. 5f), suggesting that its availability for ongoing replication forks is reduced when ssDNA gaps accumulate behind forks. Moreover, bolstering CHAF1A levels by using a doxycycline-inducible overexpression system completely rescued fork stability in cells depleted of BRCA1 and BRCA2 (Fig. 5g; Supplementary Fig. 5b). Taken together, these findings suggest that the retention of CAF-1 at ssDNA gaps behind the replication fork reduces its availability at ongoing replication forks, causing impaired nucleosome assembly which drives fork degradation in BRCA-deficient cells.

## Lagging strand gaps cause CAF-1 recycling defects

PRIMPOL-mediated repriming has recently been shown to promote ssDNA gap accumulation in BRCA-deficient cells[28,31,32,35,36,56]. We therefore asked if PRIMPOL activity could potentially drive HU-induced gap accumulation in BRCA-deficient cells. We employed the BrdU alkaline comet assay to measure the accumulation of replication-associated ssDNA gaps[27,57]. Importantly, the BrdU alkaline comet assay can detect ssDNA gaps in situations where only one of the complementary

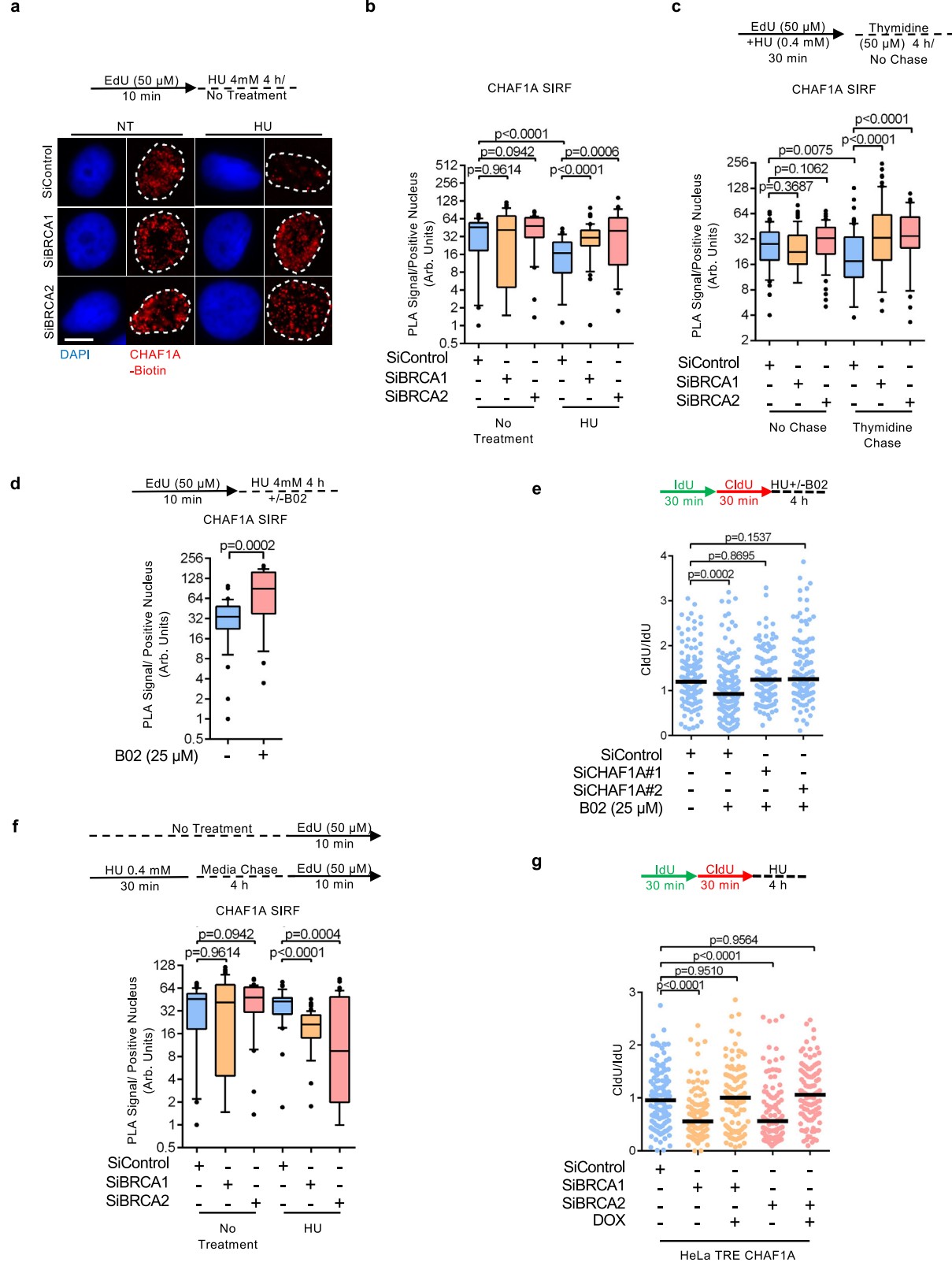

nascent DNA strands bears gaps, as opposed to the previously described S1 nuclease-based approach, where the readout necessitates the simultaneous presence of leading and lagging strand gaps. HeLa-BRCA2KO cells labeled with BrdU in the presence of a low dose of HU exhibited a greater tail moment compared to wild-type HeLa cells, indicating the presence of more ssDNA gaps (Fig. 6a, b). In line with previous findings, depletion of PRIMPOL with two separate siRNAs completely rescued ssDNA gap formation in HeLa-BRCA2KO cells under

a low dose of HU as detected by measuring DNA fibers treated with S1 nuclease (Supplementary Fig. 6a, b). Notably, we did not observe differences in replication-associated ssDNA gaps in HeLa-BRCA2KO cells under unperturbed DNA replication conditions. Importantly, in contrast to the S1 nuclease assay, the BrdU alkaline comet assay showed only a partial rescue of HU-induced ssDNA gaps by PRIMPOL depletion in HeLa-BRCA2KO cells (Fig. 6a, b; Supplementary Fig. 6a). Due to the continuous nature of DNA synthesis on the leading strand, it was

**Fig. 5 | CAF-1 recycling defects underlie fork degradation in BRCA-deficient cells. a, b** SIRF assay showing that CHAF1A unloading from nascent DNA upon replication fork arrest is deficient in BRCA1 or BRCA2-depleted HeLa cells. To induce fork arrest, cells were treated with 4 mM HU after EdU labeling. Representative micrographs (scale bar represents 10 μm) (**a**) and quantifications (**b**) are shown. At least 30 positive cells were quantified for each condition. Center line indicates the median, bounds of box indicate the first and third quartile, and whiskers indicate the 10th and 90th percentile. The *p*-values (Mann–Whitney test, two-tailed) are listed at the top. A schematic representation of the SIRF assay conditions is also presented. **c** SIRF assay showing that, in BRCA1 or BRCA2-depleted HeLa cells, CHAF1A is retained on nascent DNA behind the replication fork after recovery from replication stress. Cells were labeled with EdU in the presence of low-dose HU (0.4 mM for 30 min) to induce replication stress, washed, and chased for 4 h in fresh media containing 50 μM thymidine. At least 35 positive cells were quantified for each condition. Center line indicates the median, bounds of box indicate the first and third quartile, and whiskers indicate the 10th and 90th percentile. The *p*-values (Mann–Whitney test, two-tailed) are listed at the top. A schematic representation of the SIRF assay conditions is also presented. **d** SIRF assay showing that simultaneous treatment with 25 μM B02 and 4 mM HU elicits CHAF1A unloading defects in HeLa cells. At least 25 positive cells were quantified for each condition. Center line indicates the median, bounds of box indicate the first and third quartile, and whiskers indicate the 10th and 90th percentile. The *p*-values (Mann–Whitney test, two-tailed) are listed at the top. A schematic

representation of the SIRF assay conditions is also presented. **e** DNA fiber combing assay showing that CHAF1A knockdown restores fork protection to wild-type HeLa cells treated with the RAD51 inhibitor B02. The ratio of CldU to IdU tract lengths is presented, with the median values marked on the graph. The *p*-values (Mann–Whitney test, two-tailed) are listed at the top. A schematic representation of the DNA fiber combing assay conditions is also presented. **f** SIRF assay showing that prior replication stress reduces the levels of CHAF1A at ongoing replication forks in BRCA1 or BRCA2-depleted HeLa cells. Cells were subjected to low-dose HU (0.4 mM for 30 mins) to induced replication stress, chased in fresh media for 4 h to recover from replication stress, then labeled with EdU. At least 40 positive cells were quantified for each condition. Center line indicates the median, bounds of box indicate the first and third quartile, and whiskers indicate the 10th and 90th percentile. The *p*-values (Mann–Whitney test, two-tailed) are listed at the top. A schematic representation of the SIRF assay conditions is also presented. **g** DNA fiber combing assay showing that CHAF1A overexpression suppresses HU-induced fork degradation in BRCA1 or BRCA2-depleted HeLa cells. CHAF1A expression is under the control of the tetracycline responsive element (TRE), and is induced upon doxycycline (DOX) treatment. The ratio of CldU to IdU tract lengths is presented, with the median values marked on the graph. The *p*-values (Mann–Whitney test, two-tailed) are listed at the top. A schematic representation of the DNA fiber combing assay conditions is also presented. Western blots showing CHAF1A overexpression are presented in Supplementary Fig. 5b.

---

previously suggested that PRIMPOL likely serves as a dedicated leading strand primase during replication stress[58]. We thus hypothesized that PRIMPOL-independent gaps may in part be explained by gaps on the lagging strand which form in a Polα-dependent manner.

Unligated Okazaki fragments result in the chromatin accumulation of poly(ADP-ribose) (PAR) chains in S-phase, which can be detected upon inhibition of the poly(ADP-ribose) glycohydrolase (PARG) enzyme[59]. Indeed, SIRF experiments on cells subjected to PARG inhibition (PARGi) showed that depletion of the OF ligase LIG1 in wild-type cells results in increased PAR chain formation (Supplementary Fig. 6c, d). We reasoned that, when gaps occur on the lagging strand, the nicked DNA structure which is the substrate of LIG1 during OF ligation is not formed, since DNA synthesis on the OF is not completed. Thus, LIG1 depletion should increase PAR chain signal in cells with completed OF synthesis, but not in cells which accumulate gaps precluding nick formation. Under normal conditions, PAR SIRF experiments showed no difference in PAR chromatin levels in BRCA-proficient and BRCA1/2-depleted cells (Supplementary Fig. 6c, d). LIG1 depletion yielded a detectable increase in PAR SIRF signal, but failed to elicit differences between BRCA-proficient and BRCA1/2-depleted cells, suggesting that Okazaki fragment synthesis remains largely unperturbed in BRCA-deficient cells under normal growth conditions. We next performed SIRF to detect chromatin PAR chains after subjecting cells to EdU labeling under a low dose of HU. Interestingly, HU-induced replication stress resulted in a modest reduction in SIRF signal in BRCA1/2-depleted cells which was drastically exacerbated upon the depletion of LIG1 (Fig. 6c, d). This suggests that BRCA-deficient cells accumulate incompletely synthesized Okazaki fragments due to lagging strand gap formation upon encountering HU-induced replication stress.

We next sought to directly test if frequent Polα-mediated repriming during transient HU-induced replication stress prior to fork stalling, could drive lagging strand gap accumulation and CAF-1 recycling defects in BRCA-deficient cells. The retinoid ST1926 was previously shown to abolish Polα activity resulting in replication fork uncoupling at sufficiently high doses[60]. Indeed, treatment of HeLa cells with 10 μM ST1926 induced maximal chromatin-bound RPA levels within 5 min (Supplementary Fig. 6e, f), indicating a robust and immediate inhibition of Polα activity. We next performed SIRF on EdU-labeled cells subjected to a high dose of HU in the presence of ST1926. Strikingly, Polα inhibition, while having a minimal impact on BRCA-proficient cells, completely suppressed CHAF1A retention at stalled

forks in BRCA1 and BRCA2-depleted cells (Fig. 6e, f). In contrast, PRIMPOL depletion did not affect this retention (Supplementary Fig. 6g). These results suggest that Polα-mediated lagging strand repriming during replication stress is responsible for the CHAF1A recycling defects in BRCA-deficient cells.

Replication-associated ssDNA gaps are likely to be immediately coated by the RPA complex. Interestingly, the recruitment of HIRA to DNA during transcription has been shown to be dependent on RPA1[61]. We wondered if RPA-coated ssDNA could account for the increased presence of HIRA observed at stalled forks in BRCA-deficient cells. Indeed, depletion of RPA1 restored HIRA at stalled forks to similar levels in both BRCA-proficient and BRCA1/2-depleted cells (Fig. 6g, h; Supplementary Fig. 6h). Taken together, these results suggest that the prevalence of lagging strand gaps, caused by Polα-dependent repriming, drives the aberrant retention of CAF-1 at stressed replication forks in BRCA-deficient cells. Furthermore, RPA complexes coating ssDNA at these gaps recruit HIRA in the proximity of HU-arrested forks in BRCA-deficient cells.

## PCNA unloading ensures CAF-1 mediated fork protection

We previously showed that PCNA-K164R cells exhibit PCNA-unloading defects owing to their inability to mitigate lagging strand ssDNA gaps, therefore interfering with CAF-1 recycling at replication forks[27]. Interestingly, similar to BRCA-deficient cells, CHAF1A depletion also rescued fork stability in PCNA-K164R cells (Supplementary Fig. 7a, b). Thus, we sought to test if PCNA unloading defects cause the CAF-1 retention at stalled forks observed in BRCA-deficient cells. SIRF experiments revealed that loss of ATAD5 enhanced CHAF1A retention on nascent DNA in wild-type cells and caused fork degradation, but did not exacerbate CHAF1A retention and fork degradation in BRCA-deficient cells (Supplementary Fig. 7c, d). Similar results were obtained for LIG1, while LIG3 did not differentially affect CHAF1A retention or fork protection (Supplementary Fig. 7c–e). Importantly, similar to what we observed for CHAF1A, SIRF experiments revealed an abnormal retention of PCNA at stalled replication forks in BRCA1/2-depleted cells upon treatment with a high dose of HU (Fig. 7a, b). Polα inhibition completely abrogated this abnormal PCNA retention, indicating that lagging strand gaps are responsible for the PCNA unloading defects in BRCA-deficient cells.

Since the results described above suggested that PCNA-CAF-1 retention at ssDNA gaps promotes fork degradation in BRCA-deficient cells, we next assessed if correcting this defective PCNA unloading

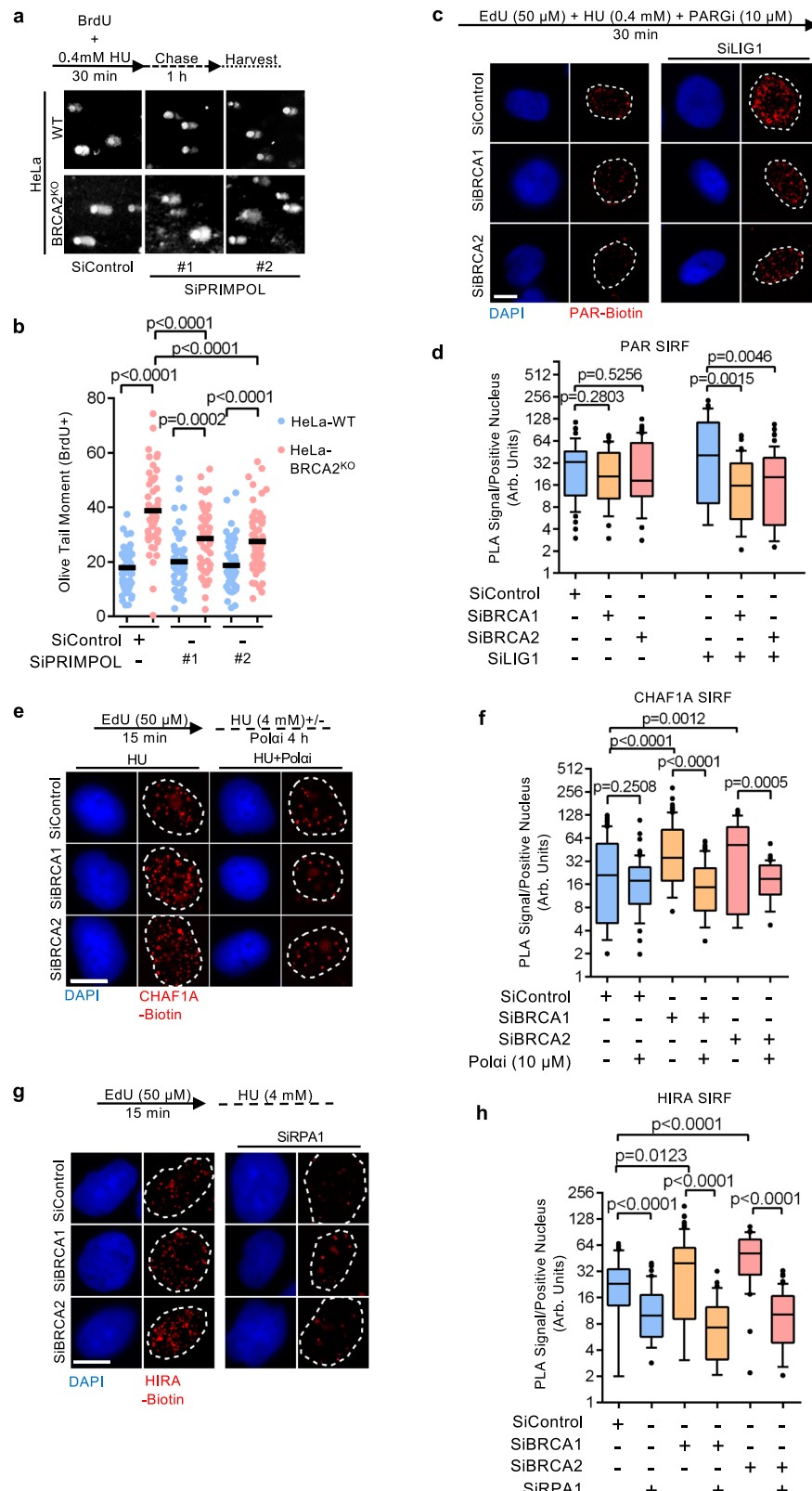

could restore fork stability to these cells. Recent work showed that the bromodomain extra-terminal (BET) family of proteins, namely BRD2, BRD3, and BRD4, form complexes with ATAD5 and inhibit its PCNA unloading activity[62,63]. We reasoned that inactivating these BET proteins would enhance PCNA removal from DNA and thus reverse the effect of PCNA unloading defects in BRCA-deficient cells. Strikingly, depleting BRD3 and BRD4 restored fork stability to HeLa-BRCA2$^{KO}$ cells

(Fig. 7c; Supplementary Fig. 7f). Importantly, co-depletion of CHAF1A with either BRD3 or BRD4 restored nascent fork degradation to HeLa-BRCA2$^{KO}$ cells, suggesting that correction of PCNA unloading defects in BRCA-deficient cells restores fork stability in a CAF-1-dependent manner.

To more directly assess if PCNA unloading defects cause fork degradation in BRCA-deficient cells, we next created PCNA mutants

**Fig. 6 | Accumulation of lagging strand gaps in BRCA-deficient cells. a, b** BrdU alkaline comet assay showing that PRIMPOL depletion reduces, but does not abolish replication stress-induced ssDNA accumulation in BRCA2-knockout HeLa cells, indicating the presence of PRIMPOL-independent gaps. Representative micrographs (**a**) and quantifications (**b**) are shown. At least 45 nuclei were quantified for each condition. The mean values are represented on the graph, and the *p*-values (*t*-test, two-tailed, unpaired) are listed at the top. A schematic representation of the assay conditions is also presented. RT-qPCR experiments confirming PRIMPOL depletion are presented in Supplementary Fig. 6a. **c, d** SIRF assay showing that, upon replication stress, LIG1 knockdown induces PAR chain formation in wild-type HeLa cells, but not in BRCA1 or BRCA2-depleted HeLa cells upon treatment with a low dose of HU (0.4 mM), indicating the prevalence of lagging strand gaps in these cells. Representative micrographs (scale bar represents 10 μm) (**c**) and quantifications (**d**) are shown. At least 35 positive cells were quantified for each condition. Center line indicates the median, bounds of box indicate the first and third quartile, and whiskers indicate the 10th and 90th percentile. The *p*-values (Mann–Whitney test, two-tailed) are listed at the top. A schematic representation of the SIRF assay

conditions is also presented. Western blots confirming the co-depletion are shown in Supplementary Fig. 6d. **e, f** SIRF assay showing that Polα inhibition suppresses the increased CHAF1A retention in BRCA1 or BRCA2-depleted HeLa cells. Representative micrographs (scale bar represents 10 μm) (**e**) and quantifications (**f**) are shown. At least 35 positive cells were quantified for each condition. Center line indicates the median, bounds of box indicate the first and third quartile, and whiskers indicate the 10th and 90th percentile. The *p*-values (Mann–Whitney test, two-tailed) are listed at the top. A schematic representation of the SIRF assay conditions is also presented. **g, h** SIRF assay showing that RPA1 co-depletion restores HIRA levels to the same levels in wild-type and BRCA1 or BRCA2-knockodown HeLa cells. Representative micrographs (scale bar represents 10 μm) (**g**) and quantifications (**h**) are shown. At least 30 positive cells were quantified for each condition. Center line indicates the median, bounds of box indicate the first and third quartile, and whiskers indicate the 10th and 90th percentile. The *p*-values (Mann–Whitney test, two-tailed) are listed at the top. A schematic representation of the SIRF assay conditions is also presented. Western blots confirming the co-depletion are shown in Supplementary Fig. 6h.

unable to accumulate on chromatin. We previously obtained a clonal line in pseudotriploid human 293T cells bearing genetic knockouts of four out of five endogenous PCNA alleles, with the final allele encoding a PCNA-K164R mutation (293T-PCNA$^{K164R(hyp)}$)[27]. Importantly, we showed that these cells exhibit lower endogenous PCNA expression and the stable complementation with wild-type PCNA restored near-wild-type characteristics to these cells, thus establishing their suitability for complementation with potential PCNA mutants. Studies in *S. cerevisiae* have characterized point mutations on PCNA's trimer interface that interfere with stable homotrimer formation[21,64–66]. We generated two of the corresponding mutations, namely PCNA-C81R and PCNA-D150E, in human PCNA (Supplementary Fig. 7g). Using a lentiviral expression system, we stably complemented 293T-PCNA$^{K164R(hyp)}$ cells with either PCNA-WT, PCNA-C81R or PCNA-D150E variants (Supplementary Fig. 7h). We next tested if these mutations could ameliorate PCNA unloading defects caused by ATAD5 or LIG1 depletion. As expected, ATAD5 depletion increased chromatin-bound PCNA in 293T-PCNA$^{WT}$ cells, however, 293T-PCNA$^{C81R}$ and 293T-PCNA$^{D150E}$ cells exhibited no increase in PCNA chromatin association upon ATAD5 depletion (Fig. 7d). Similarly, LIG1 depletion also increased chromatin-bound PCNA levels in 293T-PCNA$^{WT}$ cells, in line with Okazaki fragment ligation being a prerequisite for PCNA unloading[21]. In contrast, LIG1 knockdown failed to cause PCNA chromatin retention in 293T-PCNA$^{C81R}$ and 293T-PCNA$^{D150E}$ cells (Fig. 7e). These results confirm that PCNA-C81R and PCNA-D150E mutants have reduced chromatin retention and can correct PCNA unloading defects in human cells. We next assessed if PCNA-C81R and PCNA-D150E can ameliorate fork degradation in BRCA-deficient cells. Indeed, while both BRCA2 depletion as well as RAD51 inhibition using B02 elicited fork degradation in 293T-PCNA$^{WT}$ cells, fork stability remained intact in 293T-PCNA$^{C81R}$ and 293T-PCNA$^{D150E}$ cells (Fig. 7f, g; Supplementary Fig. 7i). Importantly, while CHAF1A depletion restored fork stability to B02-treated 293T-PCNA$^{WT}$ cells, it caused fork degradation in B02-treated 293T-PCNA$^{C81R}$ and 293T-PCNA$^{D150E}$ cells (Fig. 7g; Supplementary Fig. 7j). These results suggest that, in cells with impaired BRCA/RAD51 function, the restoration of PCNA unloading rescues fork stability by reinstating CAF-1 function at replication forks. Taken together, these observations imply that CAF-1 is a direct effector of the BRCA/RAD51 pathway of fork protection whose function is ensured by lagging strand gap suppression and efficient PCNA unloading (Fig. 7h; Supplementary Fig. 8a).

## Discussion
### Nucleosome assembly governs BRCA-mediated fork stability
Replication-coupled nucleosome assembly is mediated by CAF-1, which operates at replication forks in a PCNA-dependent manner[14,15]. CAF-1-mediated nucleosome assembly depends on its interaction with ASF1, which packages H3/H4 heterodimers

subsequently used by CAF-1 to assemble (H3/H4)$_2$ tetramers on DNA[67–70]. CAF-1 preferentially interacts with the canonical histone isoform H3.1[50]. ASF1 also participates in replication-independent deposition of histone H3.3, by cooperating with the histone chaperone HIRA[50,71]. In humans, two ASF1 paralogs exist: ASF1A and ASF1B. Of the two, only ASF1A is capable of interacting with both CAF-1 and HIRA, and appears to do so in a mutually exclusive manner[50,72,73].

The presence of nucleosomes has been shown to act as a barrier to DNA resection by nucleases[74]. We previously demonstrated that CAF-1 is essential for suppression of fork degradation[27]. In the present study, we show that ASF1A inactivation is epistatic with BRCA1/2 deficiency in eliciting fork degradation (Fig. 3a). Moreover, restoring CAF-1 function at forks, through either CHAF1A overexpression or by enhancing PCNA unloading, rescued fork stability in BRCA-deficient cells (Figs. 5g and 7). Last, fork rescue upon CHAF1A inactivation in BRCA-deficient cells depended on the ASF1A-HIRA-H3.3 pathway of replication-independent nucleosome deposition (Figs. 3b and 4e). These findings unveil nucleosome assembly as an essential effector of BRCA-mediated replication fork protection. Importantly, our findings suggest that this process is clinically relevant, since we find that CHAF1A inactivation in BRCA-deficient cells promotes chemoresistance.

Recently published work revealed a role for CAF-1 and ASF1-dependent nucleosome assembly in promoting HR by mediating RAD51 recruitment to ssDNA through MMS22L-TONSL[75]. It is therefore possible that the inactivation of CAF-1 and ASF1 directly results in fork instability by impairing RAD51-mediated fork protection. In the present work, we reveal that fork stability in BRCA-deficient cells upon CHAF1A inactivation occurs in a manner dependent on ASF1A, HIRA, and H3.3. Importantly, treatment with the RAD51 inhibitor B02 fails to elicit fork degradation under these conditions, suggesting that HIRA-mediated restoration of fork stability occurs in a RAD51-independent manner. These observations suggest that, in the context of replication stress, nucleosome assembly acts as the major determinant of fork stability and operates independently of the role of RAD51 in mediating HR.

Our observations indicate that the fork protective activity of HIRA only operates when CAF-1 and the BRCA pathway are simultaneously inactivated (Fig. 4e; Supplementary Fig. 4e–h). Previous studies have revealed a compensatory role of HIRA in filling nucleosome gaps on the genome in the absence of the CAF-1[52]. However, the inactivation of CHAF1A in BRCA-proficient cells does not trigger HIRA-mediated fork protection (Fig. 1; Supplementary Fig. 1). We speculate that this selective activation of HIRA is caused by its recruitment to RPA-coated ssDNA accumulating in BRCA-deficient cells, proximal to stalled forks (Supplementary Fig. 8b). Nevertheless, the activation of HIRA-mediated fork protection still

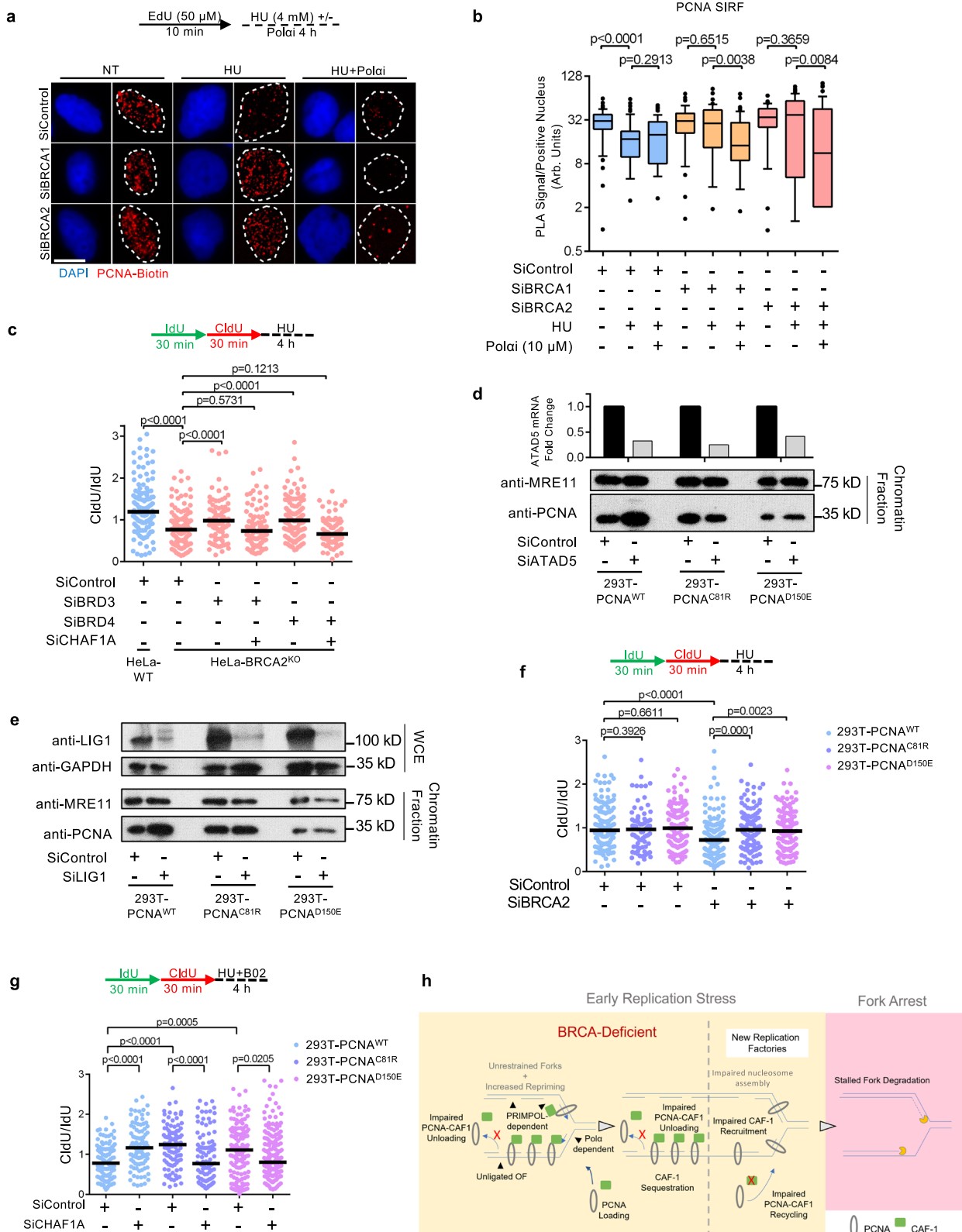

necessitates the inactivation of CHAF1A. A possible explanation for this could be the mutual exclusivity of ASF1A interactions with CAF-1 and HIRA. Indeed, ASF1 has previously been shown to ensure the supply of S-phase (H3.1-containing) histones during replication stress[76]. Therefore, it is possible that the absence of CAF-1 relieves ASF1A of its S-phase histone buffering constraints, enabling it to participate in H3.3-dependent nucleosome assembly

with HIRA. Furthermore, the RPA-dependent accumulation of HIRA at stalled forks in BRCA-deficient cell may prime HIRA to mediate efficient nucleosome assembly thereby preventing fork degradation upon CAF-1 loss (Supplementary Fig. 8b). In line with this, we observed an increased interaction between HIRA and ASF1A upon CAF-1 and BRCA inactivation during replication stress (Fig. 4g).

**Fig. 7 | PCNA unloading defects are responsible for fork degradation in BRCA-deficient cells. a, b** SIRF assay showing deficient PCNA unloading from nascent DNA upon replication fork arrest in BRCA-depleted HeLa cells. Polα inhibition by treatment with 10 μM ST1926 suppresses this defect, indicating that this PCNA retention on nascent DNA occurs at lagging strand gaps. Representative micrographs (scale bar represents 10 μm) (**a**) and quantifications (**b**) are shown. At least 50 positive cells were quantified for each condition. Center line indicates the median, bounds of box indicate the first and third quartile, and whiskers indicate the 10th and 90th percentile. The *p*-values (Mann–Whitney test, two-tailed) are listed at the top. A schematic representation of the SIRF assay conditions is also presented. **c** DNA fiber combing assay showing that co-depletion of BRD3 or BRD4 restores HU-induced fork degradation in CHAF1A-knockdown HeLa-BRCA2^KO cells. The ratio of CldU to IdU tract lengths is presented, with the median values marked on the graph. The *p*-values (Mann–Whitney test, two-tailed) are listed at the top. A schematic representation of the DNA fiber combing assay conditions is also presented. Western blots confirming the co-depletion are shown in Supplementary Fig. 7f. **d, e** Chromatin fractionation experiments showing that ATAD5 (**d**) or LIG1 (**e**) knockdown increases the chromatin levels of wild-type PCNA, but not those of PCNA C81R and D150E variants. MRE11 is used as a control for the chromatin fraction. LIG1 depletion is confirmed by western blot. ATAD5 depletion is confirmed by RT-qPCR, since no antibody was available to use to verify depletion by western blot. The average of two technical replicates is shown. **f** DNA fiber combing assay showing that BRCA2 knockdown causes HU-induced fork degradation in 293T cells

expressing wild-type PCNA, but not in 293T cells expressing PCNA variants with deficient chromatin retention. The ratio of CldU to IdU tract lengths is presented, with the median values marked on the graph. The *p*-values (Mann–Whitney test, two-tailed) are listed at the top. A schematic representation of the DNA fiber combing assay conditions is also presented. Western blots confirming the knockdown are shown in Supplementary Fig. 7i. **g** DNA fiber combing assay showing that CHAF1A knockdown suppresses HU-induced fork degradation caused by B02 treatment in 293T cells expressing wild-type PCNA, but not in 293T cells expressing PCNA variants with deficient chromatin retention. The ratio of CldU to IdU tract lengths is presented, with the median values marked on the graph. The *p*-values (Mann–Whitney test, two-tailed) are listed at the top. A schematic representation of the DNA fiber combing assay conditions is also presented. Western blots confirming the knockdown are shown in Supplementary Fig. 7j. **h** Schematic model outlining the proposed mechanism for fork degradation in BRCA-deficient cells caused by PCNA unloading defects. The failure of BRCA-deficient cells to restrain fork progression during replication stress causes gap accumulation due to repriming by PRIMPOL (on the leading strand) and Polα (on the lagging strand). Polα-mediated repriming results in formation of lagging strand gaps which preclude PCNA unloading. The persistence of PCNA behind replication forks sequesters CAF-1 away from active replication factories, thus interfering with nucleosome assembly at forks and priming them for degradation upon stalling. A more detailed version of this model figure is provided in Supplementary Fig. 8.

## BRCA-deficient cells accumulate lagging strand ssDNA gaps

Recent studies have shown that, in BRCA-deficient cells, unrestrained replication fork progression during replication stress drives ssDNA gap accumulation through excessive PRIMPOL-dependent repriming, necessitating the engagement of post-replicative gap repair pathways, such as translesion synthesis, to avoid cellular toxicity[27–32,36,37,56]. In these studies, the main experimental approach employed to detect replication-associated gaps involves measuring the shortening of DNA tracts upon treatment with the S1 nuclease, which specifically digests ssDNA substrates[77]. In this assay, for a detectable shortening of replication tracts upon S1 treatment, gaps must be present on both the leading and the lagging strands. In this study, we show that, while PRIMPOL depletion in HeLa-BRCA2^KO cells completely abolishes replication tract shortening upon treatment with the S1 nuclease, it only partially suppresses ssDNA gaps detected by the BrdU alkaline comet assay. This raises the question of the mechanism by which PRIMPOL-independent gaps occur. Recent work in reconstituted eukaryotic DNA replication systems revealed that fork progression is disproportionately impeded by leading strand obstacles, while lagging strand obstacles are efficiently bypassed via inherently frequent Polα-mediated repriming[78,79]. Repriming by PRIMPOL is thus likely to occur mostly on the leading strand, giving rise to leading strand ssDNA gaps[58]. This raises the possibility that the PRIMPOL-independent ssDNA gaps detected in BRCA2-deficient cells by BrdU alkaline comet assays could, in part, be Polα-dependent lagging strand gaps (Fig. 6a, b). Indeed, taking advantage of the fact that S-phase chromatin-bound PAR chains specifically form at fully synthesized but unligated OFs[59], we demonstrate that BRCA-deficient cells accumulate lagging strand discontinuities during replication stress (Fig. 6c, d). Recent evidence suggests that the BRCA pathway participates in the repair of post-replicative gaps[30,36,37], suggesting that BRCA-mediated gap filling could also safeguard against lagging strand gap accumulation. Indeed, in previous work, we showed that the accumulation of ssDNA gaps in the absence of PCNA ubiquitination impairs the completion of lagging strand DNA synthesis, and necessitates BRCA-mediated gap repair[27].

The unloading of PCNA from lagging strands can only occur upon OF ligation[21]. Importantly, we and others have shown that loss of PCNA ubiquitination, which ensures efficient lagging strand synthesis, results in PCNA unloading defects[27,55]. Thus, PCNA unloading represents a surrogate marker for OF synthesis and ligation defects. In this study, using SIRF assays, we show that PCNA unloading is defective upon replication stress in BRCA-deficient cells, but can be ameliorated by

Polα inhibition. While our study does not include additional approaches such as iPOND to investigate nascent DNA binding, these findings nevertheless support the notion that incompletely synthesized OFs accumulate in BRCA-deficient cells (Fig. 7a, b). Indeed, recent studies have also reported OF processing defects in BRCA-deficient cells[37,38]. In conclusion, in combination with previous studies, we provide compelling evidence that BRCA-deficient cells accumulate lagging strand gaps during replication stress.

## PCNA unloading connects gap suppression to fork stability

In the past decade, the inability to protect stalled forks from degradation by nucleases has emerged as a defining hallmark of BRCA-deficiency[6,7,9,12,46]. A direct consequence of fork degradation are gross chromosomal aberrations, which likely play a crucial role in enabling tumorigenesis and cancer evolution. Furthermore, fork degradation is also associated with the hypersensitivity of BRCA-deficient cancers to chemotherapeutic agents such as cisplatin[12]. Conversely, the restoration of fork stability drives chemoresistance in the absence of BRCA function[12,39,80]. Mechanistically, fork degradation has been shown to result in the formation of a novel substrate for endonucleolytic cleavage by MUS81, thereby enabling the restart of forks through a break-induced replication process[11]. However, despite having profound implications in driving genome instability in BRCA-deficient cells, fork degradation only partially explains PARPi sensitivity[12,30,37,39,41]. Instead, the inability to suppress replication-associated gaps in BRCA-deficient cells has recently emerged as a major predictor of PARPi sensitivity[27,28,30–32,34,37,56,81]. Nonetheless, how exactly ssDNA gaps underpin genome instability remains unclear. Recent evidence suggests that ssDNA gaps in BRCA-deficient cells may persist through mitosis into subsequent cell cycles, eliciting DNA damage when encountered by replication forks[31,36,82]. In the present study, we reveal fork degradation as a direct consequence of replication-associated gap accumulation in BRCA-deficient cells. We find that the failure to restrain forks during replication stress contributes to the accumulation of lagging strand ssDNA gaps, which interferes with PCNA unloading and CAF-1 recycling in BRCA-deficient cells (Fig. 7h; Supplementary Fig. 8a). We moreover show that the reduction in CAF-1 availability at replication forks drives their degradation due to impaired replication-associated nucleosome assembly in BRCA-deficient cells.

Traditional models of BRCA-mediated fork protection assert the role of BRCA-dependent RAD51 nucleofilament assembly on reversed

fork arms in acting as a direct obstacle to the action of nucleases[6,7,34]. However, recent evidence suggests that RAD51 assembly at replication forks may occur at gaps proximal to the replication forks undergoing PRIMPOL-mediated repriming[35]. Additional evidence suggests a role for the BRCA pathway in orchestrating RAD51-dependent replication fork slowing and reversal[29]. Critically, fork reversal mechanisms have been shown to play an essential role in limiting PRIMPOL-mediated repriming during replication stress[32,36,56,83]. Given that we now describe a role for replication-associated gap formation in driving fork degradation, we speculate that a major mechanism by which the BRCA pathway orchestrates RAD51-mediated fork protection is by ensuring the engagement of RAD51 at ssDNA gaps proximal to, as well as behind, stressed replication forks. This enables RAD51 to suppress de novo ssDNA gap formation as well as rapidly repair post-replicative gaps, thereby ensuring timely PCNA unloading and CAF-1-dependent nucleosome assembly, which underlie fork protection. In conclusion, we propose a model which unifies gap suppression and fork protection, two critical functions of the BRCA pathway, and connects them to the fundamental replicative function of PCNA in orchestrating nucleosome assembly at replication forks (Fig. 7h; Supplementary Fig. 8).

## Methods

### Cell culture and protein techniques
Human 293T, RPE1, and HeLa cells were grown in DMEM supplemented with 10% Fetal Calf Serum. For CHAF1A gene knockout, the commercially available CHAF1A CRISPR/Cas9 KO plasmid was used (Santa Cruz Biotechnology sc-402472). Transfected HeLa or 293T cells were FACS-sorted into 96-well plates using a BD FACSAria II instrument. Resulting colonies were screened by western blot. The BRCA2-knockout HeLa cells were created in our laboratory and were previously described[40]. RPE1-p53$^{KO}$-BRCA1$^{KO}$ were obtained from Dr. Alan D'Andrea (Dana-Farber Cancer Institute, Boston, MA)[84]. 293T cells with hypomorph PCNA expression were created in our laboratory and previously described[27]. For exogenous PCNA expression, pLV[Exp]-Puro-CMV lentiviral constructs encoding wild type or the indicated variants were obtained from Cyagen. For doxycycline-induced CHAF1A overexpression, the pLV[Exp]-Puro-TRE > hCHAF1A lentiviral construct (Cyagen) was used. Infected cells were selected by puromycin.

Gene knockdown was performed using Lipofectamine RNAiMAX. AllStars Negative Control siRNA (Qiagen 1027281) was used as control. The following oligonucleotide sequences (Stealth or SilencerSelect siRNA, ThermoFisher; unless otherwise noted) were used: BRCA1: AATGAGTCCAGTTTCGTTGCCTCTG; BRCA2: GAGAGGCCTGTAAA GACCTTGAATT; SMARCAL1: CACCCTTTGCTAACCCAACTCATAA; ZRANB3: TGGCAATGTAGTCTCTGCACCTATA; ATAD5: GGTACGCTTT AAGACAGTTACTGTT; E2F7: GGACGATGCATTTACAGATTCTCTA; CHAF1A#1: s19499; CHAF1A#2: HSS115231; PRIMPOL #1: GAGGAA ACCGTTGTCCTCAGTGTAT (Horizon Discovery); PRIMPOL #2: 39536 ASF1A#1: CAGAGAGCAGTAATCCAAATCTACA; ASF1A#2: s226043; HIRA#1: HSS111075; HIRA#2: HSS186934; DAXX: s3935; H3F3A: s51241; H3F3B: s226272; LIG1: s8173; RPA1: s12127; BRD3: s23901; BRD4: s15544; LIG3: s8177; ASF1B: s31346.

Denatured whole cell extracts were prepared by boiling cells in 100 mM Tris, 4% SDS, 0.5 M β-mercaptoethanol. Chromatin fractionation was performed by subjecting cells to extraction with 0.1% Triton X-100[85]. Antibodies used for Western blot were: CHAF1A (Cell Signaling Technology 5480); ASF1A (Santa Cruz Biotechnology sc-53171); BRCA1 (Santa Cruz Biotechnology sc-6954); BRCA2 (Calbiochem OP95); ZRANB3 (Invitrogen PA5-65143); SMARCAL1 (Invitrogen PA5-54181); HIRA (Abcam 129169); DAXX (Invitrogen PA5-79137); RPA1 (Cell Signaling Technology 2198); LIG1 (Bethyl A301-136A); MRE11 (GeneTex GTX70212); PCNA (Cell Signaling Technology 2586); ubiquitinated PCNA (Cell Signaling Technology 13439); BRD3 (Bethyl A302-368A); BRD4 (Bethyl A700-005); GAPDH (Santa Cruz Biotechnology sc-47724);

LIG3 (Santa Cruz Biotechnology sc-135883); ASF1B (Cell Signaling Technology 2769). All antibodies were used at a dilution of 1:500. Uncropped scans of all blots are provided as a Source data file.

RAD51 and γH2AX immunofluorescence[86] and RPA2 immunofluorescence[27] were performed with the following primary antibodies: RAD51 (Abcam ab133534); γH2AX (Millipore 05-636); RPA2 (Abcam ab2175). Secondary antibodies used were AlexaFluor 488 or AlexaFluor 568 (Invitrogen A11001, A11008, A11031, and A11036). Slides were imaged using a DeltaVision Elite confocal microscope. The number of foci/nucleus was quantified using Fiji (ImageJ2) software.

### Drug sensitivity assays
For clonogenic survival assays, 1000 siRNA-treated cells were seeded per well in 6-well plates and incubated with the indicated doses of cisplatin. Media was changed after 1 day and colonies were stained after 10–14 days. Colonies were washed with PBS, fixed with a solution of 10% methanol and 10% acetic acid, and stained with 1% crystal violet (Aqua solutions).

Neutral and BrdU alkaline comet assays were performed[27] using the Comet Assay Kit (Trevigen 4250-050). For the BrdU alkaline comet assay, cells were incubated with 100 μM BrdU as indicated. Slides were imaged on a Nikon microscope operating the NIS Elements V1.10.00 software. Tail moment was analyzed using CometScore 2.0.

### DNA fiber combing assays
Cells were treated with siRNA and/or drugs according to the labeling schemes presented. Cells were incubated with 100 μM IdU and 100 μM CldU as indicated. Hydroxyurea (4 mM) and additional inhibitors (50 μM mirin for MRE11 inhibition; 30 μM C5[87] for DNA2 inhibition; 25 μM B02 for RAD51 inhibition; 10 μM ST1926 for Polα inhibition) were added as indicated. Next, cells were collected and processed using the FiberPrep kit (Genomic Vision EXT-001). DNA molecules were stretched onto coverslips (Genomic Vision COV-002-RUO) using the FiberComb Molecular Combing instrument (Genomic Vision MCS-001). Slides were then stained with antibodies detecting CldU (Abcam 6236) and IdU (BD 347580) and incubated with secondary Cy3 and Cy5 antibodies (Abcam 6946 and Abcam 6565). Finally, the cells were mounted onto coverslips and imaged using a confocal microscope (Leica SP5), and analyzed using LASX 3.5.7.23225 software. At least 70 tracts were quantified for each sample unless otherwise specified.

### S1 nuclease fiber spreading assay
The S1 nuclease treatment in combination with DNA fiber spreading for the detection of replication-associated ssDNA gaps was done as described[77]. Briefly, exponentially-growing cells were labeled with 100 μM IdU and 100 μM CldU for the indicated times in the presence or absence of 0.4 mM HU. After labeling, cells were washed with PBS and permeabilized with CSK100 buffer (100 mM NaCl, 10 mM MOPS pH 7, 3 mM MgCl$_2$, 300 mM sucrose, 0.5% Triton X-100) for 10 min at room temperature. Permeabilized cells were subjected to subsequent washes with PBS and S1 nuclease buffer (30 mM sodium acetate, 10 mM zinc acetate, 5% glycerol, 50 mM NaCl, pH 4.6). Nuclei were then treated with either 20 U/ml S1 nuclease (Invitrogen 18001016) or S1 buffer without nuclease for 30 min at 37 °C. After treatment, nuclei were washed with PBS, collected in PBS with 0.1% BSA, and centrifuged at 5200 × $g$ for 5 min. The pelleted nuclei were resuspended in PBS, and 2 μL of the solution was spotted on microscopy slides, followed by lysis with 8 μL lysis buffer (200 mM Tris−HCl pH 7.5, 50 mM EDTA, 0.5% SDS) and spreading. Slides were allowed to dry, followed by fixation with methanol and acetic acid (3:1) for 5 min, followed again by drying. Dried slides were rinsed in distilled water and denatured with 2.5 M HCL for 90 min. Slides were then washed with PBS followed by staining and imaging in accordance with the DNA fiber combing protocol outlined above. At least 45 tracts were quantified for each condition.

## Quantification of gene expression by real-time quantitative PCR (RT-qPCR)

Total mRNA was purified using TRIzol reagent (Life Tech). To generate cDNA, 1 μg RNA was subjected to reverse transcription using the RevertAid Reverse Transcriptase Kit (Thermo Fisher Scientific) with oligo-dT primers. Real-time qPCR was performed with PerfeCTa SYBR Green SuperMix (Quanta), using a CFX Connect Real-Time Cycler (BioRad). The cDNA of GAPDH gene was used for normalization. Primers used were: H3F3A for: TCTGGTGCGAGAAATTGCTC; H3F3A rev: TCT TAAGCACGTTCTCCACG; H3F3B for: CGAGAGATTCGTCGTTATCAG; H3F3B rev: TGACTCTCTTAGCGTGGATG; ATAD5 for: AGGAAGA-GATCCAACCAACG; ATAD5 rev: ATGTTTCGAAGGGTTGGCAG; GAPDH for: AAATCAAGTGGGGCGATGCTG; GAPDH rev: GCAGAGATGATGACC CTTTTG; PRIMPOL for: TTCTACTGAAGTGCCGATACTGT; PRIMPOL rev: TGTGGCTTTGGAGGTTACTGA.

## Proximity ligation assays

For SIRF assays, cells were seeded in 8-well chamber slides at 50% confluency. The following day, cells were labeled with 50 μM 5-Ethynyl-2′-deoxyuridine (EdU) and treated with HU and other drugs as indicated. Cells were then extracted with 0.5% Triton X-100 in PBS for 10 min at 4 °C, followed by fixation with 4% formaldehyde in PBS for 15 min. Fixed samples were then blocked with 3% BSA in PBS at 37 °C. After blocking, samples were subjected to a Click-iT reaction with Biotin Azide for 30 min, followed by incubation with primary antibodies in 1% BSA and 0.05% Triton X-100 in PBS at 4 °C overnight. Primary Antibodies used were: Biotin (mouse: Jackson ImmunoResearch 200-002-211; rabbit: Bethyl Laboratories A150-109A); PARP1 (Cell Signaling Technology 9542); CHAF1A (Cell Signaling Technology 5480); HIRA (Abcam 129169); DAXX (Invitrogen PA5-79137); PAR (R&D systems 4335-MC-100); PCNA (Cell Signaling Technology 13110); Histone H3 (Cell Signaling Technology 4499). Following primary antibody treatment, cells were subjected to the PLA reaction using the Duolink kit (Millipore Sigma). Nuclear fluorescence signal was acquired using a DeltaVision Elite fluorescence microscope. For data analysis, cells positive for PLA fluorescence signal between biotin and the protein of interest were identified and the PLA foci were counted. To control for variabilities in EdU uptake, foci counts of each sample were normalized to the respective biotin-biotin control using the geometric mean of the PLA fluorescence signal from positive cells and represented as PLA signal.

For ASF1A-HIRA PLA assays, cells were seeded into 8-chamber slides and 24 h later, were treated with HU (4 mm for 3 h) as indicated. Cells were then permeabilized with 0.5% Triton X-100 for 10 min at 4 °C, washed with PBS, fixed at room temperature with 3% paraformaldehyde in PBS for 10 min, washed again in PBS and then blocked in Duolink blocking solution (Millipore Sigma DUO82007) for 1 h at 37 °C, and incubated overnight at 4 °C with primary antibodies. The primary antibodies used were: ASF1A (Santa Cruz Biotechnology sc-53171) and HIRA (Novus NBP3-04893). Samples were then subjected to a proximity ligation reaction using the Duolink kit (Millipore Sigma DUO92008). Slides were imaged using a DeltaVision Elite microscope with SoftWorx 6.5.2 software, and images were analyzed using Fiji (ImageJ2) software.

## Cellular gene dependency analyses

Cellular dependency data was obtained from the DepMap Public 21Q3 dataset using the DepMap portal (depmap.org/portal). Gene knockout effects (CERES) from project Achilles CRISPR screens were obtained for BRCA1-wild-type cell lines and for cell lines bearing a deleterious mutation for BRCA1 in accordance to the Cancer Cell Line Encyclopedia Project (CCLE)[88–91]. CHAF1A and HIRA gene effects were then plotted as a regression for BRCA1-wild type and BRCA-mutant samples. *P*-values for the likelihood of non-zero slopes were ascertained.

## TCGA dataset analyses

Genomic, transcriptomic, and survival data for ovarian cancer samples[92], part of The Cancer Genome Atlas (TCGA), were obtained from cBioPortal[93]. Survival datasets were sorted by BRCA2-status and all BRCA2-mutant samples were used for subsequent analyses. Samples were divided into two groups based on CHAF1A expression status in the patient tumor samples: high (0–50th percentile) and low (51st–100th percentile). Mantel-Cox log ranked *t* test was used for statistical analyses of the datasets using Prism software.

## Statistics and reproducibility

For clonogenic assays, the two-way ANOVA statistical test was used. For immunofluorescence assays (except γH2AX staining), the DNA fiber assays, and the proximity ligation (SIRF) assays the Mann–Whitney statistical test was used; In addition, Kruskal–Wallis with Dunn's multiple comparison analyses are presented in the Source Data file for these experiments. For γH2AX immunofluorescence and comet assays, the *t*-test (two-tailed unpaired) was used; In addition, one-way ANOVA with Holm-Sidak's multiple comparison analyses are presented in the Source Data file for these experiments. For immunofluorescence, DNA fiber combing, proximity ligation assays, and comet assays, results from one experiment are shown; the results were reproduced in at least one additional independent biological conceptual replicate. Western blot experiments were reproduced at least two times. Statistical analyses were performed using GraphPad Prism 6 and Microsoft Excel v2205 software. Statistical significance is indicated for each graph.

## Reporting summary

Further information on research design is available in the Nature Research Reporting Summary linked to this article.

## Data availability

TCGA datasets were obtained from https://www.cbioportal.org. The datasets generated and/or analyzed in the current study are available from the corresponding author upon reasonable request. Source data are provided with this paper.

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

## Acknowledgements

We would like to thank Drs. Alan D'Andrea, Binghui Shen, and Boris Pfander for materials and advice; and the Penn State College of Medi-cine Flow Cytometry and Imaging cores. This work was supported by the National Institutes of Health (R01ES026184 and R01GM134681 to G.L.M.; R01CA244417 to C.M.N.).

## Author contributions

T.T. and G.L.M. designed the experiments. T.T., A.D., J.S., E.M.S., and C.M.N. conducted the experiments. T.T. and G.L.M. wrote the paper.

## Competing interests

The authors declare no competing interests.
