## [Peer Review File · Nature Communications]

Lagging strand gap suppression connects BRCA-mediated fork protection to nucleosome assembly through PCNA-dependent CAF-1 recyclingREVIEWER COMMENTS

Reviewer #1 (Remarks to the Author):

The article by Thakar et al aims to study the link between ssDNA gaps, BRCA-mediated fork protection and nucleosome assembly during DNA replication. The authors first observe that that CAF-1 inactivation rescues fork stability in BRCA depleted cells, based on the CIdU/IdU incorporation ratio in fiber assays. From this, they build the hypothesis that nucleosome assembly by CAF-1 contributes to BRCA-dependent mechanisms at replication forks. They then test 1) how depletion/inhibition of other factors affect this interplay using fiber assays (RAD51, MRN complex, ASF1A, HIRA, etc), 2) how proximity of different proteins (CHAF1A, HIRA, PCNA, etc) to replicated DNA is affected using PLA-based readouts. Using these experiments, which constitute the large bulk of the data together with few additional phenotypic assays (i.e. comet assays), the author aim to derive a complex network of relationships between nucleosome assembly and DNA replication fork protection factors.

Understanding these relationships is an important problem in biology, and some of the presented findings may contribute to future studies in this direction. However, in the current manuscript, the experimental data is largely correlative and limited in providing direct dependencies within this network. The logic behind the flow of experiments is at times very hard to follow, and the conclusions are in my opinion largely overstated, with the mechanistic claims unjustified. Some specific concerns are listed below.

1) The structure of data presentation should be corrected.

a. Control conditions are scattered in different figures. For example: the siCHAF1A+B02 in WT cells condition that should be an important control for Figure 1G-H is shown in Figure 5D and not discussed in the context of Figure 1 (and in the description of Figure 5D there is no mention of Figure 1G-H either). These are essentially part of the same experiment. In this specific example, the observation that B02 treatment already rescues the siCHAF1A effect is important for the argumentation used in the related text on page 8, as it implies that RAD51 is involved in this effect also in WT cells (on the contrary to the statement that says that this is "RAD51-independent" on page 8). How would B02 treatment affect CIdU/IdU ratios in BRCA1/2KO cells? This is also an important control missing from this set of experiments.

In general, this makes the conclusion weaker because selected data is used to draw such conclusions, and the real dependencies of the measured readout are unclear.

b. Data points are used in multiple figures. siCTRL and siCHAF1A (in BRCA2KO) datasets in Figure 3B appear to be the same used in Figure 4E for the same conditions. This should be clarified/corrected. The same goes for the siCTRL and siCHAF1A (in Hela-WT) for Figures 3C and 3D, etc.

c. Controls are missing. Some examples are:

-It is known that CHAF1A depletion affect S phase progression, therefore fibers should have different length in this condition. The authors do not report about this effect and do not mention if this is observed in their data. These are basic information and tests that should be included in the article. Also how is fiber length affected by the different treatments is an important factor in helping the readership to interpret the data. These measurements should be reported in all fiber assays.

-Another example is related to the controls for the PLA experiments in Figure 4C-D. To be able to trust the assay in these conditions, it should be shown how inhibiting replication affects SIRF. Moreover, controls without HU and/or upon CAF-1 depletion should also be presented to characterize how HIRA behaves in normal replication condition and upon CAF-1 removal. This is essential in supporting the interpretation of the authors that HIRA takes over nucleosome assembly in specific situations.

-The choice of depleting ASF1A over ASF1B (the S-phase specific ASF1 variant) is odd, and ASF1B depletions should be included in this study and taken into account in the interpretations.

2) The methods (fiber assays, PLA-based readouts, comet assays) are not sufficiently powerful to demonstrate mechanistic relationships. Some examples of overstatements are here, but in my opinion virtually all statements in the abstract are not sufficiently supported by the data.

a. page 11, line 4, the author use the wording "replication-coupled DNA damage", there is no evidence in their data that the effects observed is pertinent to replication-dependent DNA damage. It is only inferred, but not directly tested/shown.

b. PLA signal indicates proximity within 40 nm, this corresponds to more than 3 nucleosomes, or 2 replisomes, etc. It basically can not be used to assess "recruitment" as the author argue throughout their manuscript. Experiments like iPOND could be used if the authors want to claim replication-dependencies and recruitment.

3) The authors often compare many (i.e. more than 2) conditions in their experiments using statistical tools that are meant to compare only 2 groups (T-test or Mann-Whitney). For example, in figure 2B a T-test is used for the statistical analysis. T-test can only be used to compare a maximum of two groups. As in this figure multiple comparisons of 6 different conditions (so 6 groups) are done, also a test for multiple comparison, like the 2-way Anova, should be performed. Therefore, we recommend that these analyses are redone with the appropriate statistical approaches.

4) The rationale for the experiments shown in Figure 7 was not clear to me. It is also not clear why these mutants lead to an increase in CIIdU/IdU ratio shown in Figure 7G (compare siCTRL for the mutants vs the WT PCNA samples). These unexpected variations again argue a limit in the robustness of the fiber assay in deriving the conclusions that the authors are making.

Reviewer #2 (Remarks to the Author):

In this manuscript, Thakar et al uncovered a role of CAF-1 mediated nucleosome assembly in BRCA1 mediated fork protection. It is known that previously that depletion of CHAF1A, the gene encoding the large subunit of CAF-1, resulted in degradation of newly synthesized DNA, in wild type cells. Similarly BRCA1 deficiency also led to fork degradation. Interestingly, they found that depletion of CHAF1A in BRCA1 cells suppresses fork degradation. They went on to show the restoration of fork degradation in BRCA1 cells is due to sequestering PCNA and CAF-1 on ssDNA gaps, and depletion of CAF-1 in BRCA1 cells activates the HIRA mediated nucleosome assembly pathway for fork protection. Overall, this is an interesting study and the conclusions are largely supported by data presented. The following concerns should be addressed/clarified before acceptance of this study.

Major concerns:

1) In the figure 1, fork protection was performed in cells treated with HU. In figure 2, the effects on fork protection on genome stability were performed in cells treated with CPT. It is not clear whether these two different drugs will have the same effects on fork degradation in BRCA1 and BRCA2 deficient cells.

2) Fig. 3A. The effects of Asf1a depletion on fork degradation in wild type and BRCA1 mutant cells need to be revisited. What are the effects of depletion of Asf1b alone and in combination with Asf1a depletion?

3) What is the effects of Asf1a and HIRA depletion on nascent fork protection in wild type cells upon depletion of CHAF1A?

4). Fig. 5. It is not clear whether the following statement "these findings suggest that the retention of CAF-1 at ssDNA gaps behind the replication fork reduces its availability at ongoing replication forks, causing impaired nucleosome assembly which drives fork degradation in BRCA-deficient cells" is true/accurate for the following reasons. HU, which can slow down replication forks, is used for all experiments to measure fork stability. It is not clear why CAF-1 is needed for ongoing replication forks without movements under HU.

- 5). It is not clear why both CHAF1A depletion and BRCA1 deficiency are needed for activation of the HIRA-H3.3 pathway.
- 6). What is the reason for the degradation of nascent DNA, but not template DNA in cells with depletion of CHAF1?

Minor concerns:

Fig. S6A and S6B, the effects of two siRNA against PRIMPOL on ssDNA gaps are very different, which is in contrast to similar effects reported in Fig. 6A-B using another assay.

Reviewer #3 (Remarks to the Author):

In this manuscript, Thakar et al. report that the sequestration of the histone chaperone CAF-1 at ssDNA gaps causes nascent DNA degradation in BRCA1/2-deficient cells. In addition, they show that loss of CAF-1 suppresses fork degradation in BRCA1/2-deficient cells as a result of compensatory activation of HIRA-dependent histone deposition. These well-presented studies reveal a novel role for histone deposition in controlling the stability of stalled replication forks in BRCA1/2-deficient cells. As such, this work is of high interest to the DNA damage, replication and repair communities. We recommend the authors to address the points below to further strengthen the findings of their manuscript.

- 1) The authors show in Figure 2 that loss of CAF-1 reduces CPT-induced genome instability and cisplatin-mediated loss of viability in BRCA1/2-deficient cells. It would be important to also determine whether CAF-1 loss reduces cisplatin-induced genome instability in BRCA1/2-deficient cells.
- 2) The authors show that LIG1 deficiency induces Okazaki fragment maturation defects in BRCA1/2-deficient cells (Figure 6D). Does LIG1 loss enhance CAF-1 retention at nascent DNA and fork degradation in BRCA1/2-deficient cells? Are similar phenotypes also observed upon LIG3 loss, which was recently shown to cause gap accumulation in BRCA1-deficient cells (Dias et al, 2021)?
- 3) Given that PRIMPOL loss partially reduces gap formation in BRCA1/2-deficient cells (Figure 6B), it would be informative to determine whether it may have an impact on CAF-1 retention at nascent DNA in BRCA1/2-deficient cells.
- 4) Since ATAD5 loss enhances PCNA levels on chromatin (Figure 7D), it would be useful to show whether it increases CAF-1 retention at nascent DNA and fork degradation in BRCA1/2-deficient cells. Can ATAD5 overexpression reduce nascent DNA degradation in BRCA1/2-deficient cells?
- 5) Page 13. "tend to dependent" should be "tend to be dependent".

Response to referees

We would like to thank the reviewers for their helpful and constructive comments, which led to a significantly improved manuscript. To address the reviewers' concerns, we are submitting a substantially revised manuscript with 16 new figure panels, as well as 1 revised figure panel and 1 figure for reviewers. Below, please find our point-by-point reply to reviewers' comments (**our responses in red font**).

Reviewer #1

The reviewer found that our work addresses “*an important problem in biology*”, but the reviewer raises a number of points of criticism. We are addressing the reviewer's comments as indicated below.

(Remarks to the Author):

The article by Thakar et al aims to study the link between ssDNA gaps, BRCA-mediated fork protection and nucleosome assembly during DNA replication. The authors first observe that that CAF-1 inactivation rescues fork stability in BRCA depleted cells, based on the CldU/IdU incorporation ratio in fiber assays. From this, they build the hypothesis that nucleosome assembly by CAF-1 contributes to BRCA-dependent mechanisms at replication forks. They then test 1) how depletion/inhibition of other factors affect this interplay using fiber assays (RAD51, MRN complex, ASF1A, HIRA, etc), 2) how proximity of different proteins (CHAF1A, HIRA, PCNA, etc) to replicated DNA is affected using PLA-based readouts. Using these experiments, which constitute the large bulk of the data together with few additional phenotypic assays (i.e. comet assays), the author aim to derive a complex network of relationships between nucleosome assembly and DNA replication fork protection factors.

Understanding these relationships is an important problem in biology, and some of the presented findings may contribute to future studies in this direction. However, in the current manuscript, the experimental data is largely correlative and limited in providing direct dependencies within this network. The logic behind the flow of experiments is at times very hard to follow, and the conclusions are in my opinion largely overstated, with the mechanistic claims unjustified.

We would like to respectfully advocate that the methods which we are using are state-of-the-art in the field, and have been used by virtually every recent high-impact publication in the field to draw mechanistic conclusions. This fact is acknowledged by the other two reviewers, who state that our conclusions are supported by the data.

Some specific concerns are listed below.

*1) The structure of data presentation should be corrected.
a. Control conditions are scattered in different figures. For example: the siCHAF1A+B02 in WT cells condition that should be an important control for Figure 1G-H is shown in Figure 5D and not discussed in the context of Figure 1 (and in the description of Figure 5D there is no mention*

of Figure 1G-H either). These are essentially part of the same experiment. In this specific example, the observation that B02 treatment already rescues the siCHAF1A effect is important for the argumentation used in the related text on page 8, as it implies that RAD51 is involved in this effect also in WT cells (on the contrary to the statement that says that this is “RAD51-independent” on page 8). How would B02 treatment affect CldU/IldU ratios in BRCA1/2KO cells? This is also an important control missing from this set of experiments. In general, this makes the conclusion weaker because selected data is used to draw such conclusions, and the real dependencies of the measured readout are unclear.

We strongly believe that our data is logically presented, and the experiments are appropriately controlled -as also acknowledged by the other reviewers.

Regarding the example provided by the reviewer, Figures 1g,h and Figures 5d,e are certainly not part of the same experiment. Figure 5D is a SIRF experiment and not a fiber experiment, but we believe that the reviewer refers to Fig. 5e. In this figure, B02 is added to wildtype cells in order to mimic BRCA-deficiency. In wildtype cells, RAD51 is involved in fork protection, and its inhibition by B02 causes fork degradation, as previously demonstrated (Taglialatela et al, *Mol. Cell* 2017, PMID: 29053959). The purpose of the experiments in Fig. 5d,e is to show that RAD51 inhibition, similar to BRCA inactivation, also causes ssDNA gaps, which sequester CAF-1 and cause fork degradation.

In contrast, in Fig. 1g,h we use B02 to show that restored RAD51 activity is not needed for the fork protection conferred by CAF-1 loss in BRCA-deficient cells. This is an important distinction, as we never claimed that RAD51 activity is not needed for fork protection in wildtype cells (this has clearly been shown to be the case, as described above). If RAD51 function is restored upon CAF-1 depletion in BRCA2-knockout cells, then we would have expected B02 treatment to restore fork degradation under these conditions (since RAD51 loading is inhibited by B02). This is not the case, thus arguing that restored RAD51 function is not involved in conferring fork stability upon CAF-1 inactivation in a BRCA-deficient context (as we mention in the manuscript). To further demonstrate this, we now provide the new **Fig. 1g** in which we include an analysis of E2F7. We previously showed that loss of E2F7 promotes fork protection in BRCA2-deficient cells through a mechanism which involves BRCA2-independent RAD51 loading on chromatin (Clements et al, *Nucleic Acids Res* 2018, PMID: 30032296). In the new Fig. 1g we now show that depletion of either CHAF1A or E2F7 restores fork protection in BRCA2-knockout cells. However, when these cells are treated with B02, fork degradation is restored in E2F7-depleted cells, but not in CHAF1A-depleted cells. These results highlight the two different mechanisms of restored fork protection: RAD51-dependent for E2F7 loss (in accordance with our previously published findings), and RAD51-independent for CAF-1 loss.

b. Data points are used in multiple figures. siCTRL and siCHAF1A (in BRCA2KO) datasets in Figure 3B appear to be the same used in Figure 4E for the same conditions. This should be clarified/corrected. The same goes for the siCTRL and siCHAF1A (in Hela-WT) for Figures 3C and 3D, etc.

It is indeed true that, in the original manuscript, in a few cases the same data plots were presented in two different figures. The reason is that multiple samples were performed within the same experiment, however for the flow of the manuscript it made more sense to present some of the samples in different parts of the manuscript, and therefore in two different figures. Thus, some of the control samples were presented in both figures. In the revised manuscript we addressed this issue by providing a new **Fig. 3b** showing the results of independent replicate

experiment, and condensing the original figures 3C and 3D into the new **Fig. 3c** since all samples were performed within the same experiment.

c. Controls are missing. Some examples are:

-It is known that CHAF1A depletion affect S phase progression, therefore fibers should have different length in this condition. The authors do not report about this effect and do not mention if this is observed in their data. These are basic information and tests that should be included in the article. Also how is fiber length affected by the different treatments is an important factor in helping the readership to interpret the data. These measurements should be reported in all fiber assays.

The DNA fiber combing experimental setup we employed (subsequent IdU, CldU, and HU treatments) is specifically designed to measure nascent strand nucleolysis by calculating the ratios of CldU to IdU tract lengths (Quinet et al, *Methods Enzymol.* 2017, PMID: 28645379). By calculating this ratio, any unspecific impact of siRNA treatment on tract lengths is removed from the analysis, allowing the specific investigation of stalled fork degradation. This is the standard assay in the field for measuring strand degradation. Nevertheless, to address the reviewer comment, we are providing in the Source Data file for Fig. 1a and Fig. 1b the measurements of the first tract, showing the fiber length in CHAF1A-depleted cells under normal conditions in HeLa and RPE1 cells. No obvious impact of CAF-1 loss is observed.

-Another example is related to the controls for the PLA experiments in Figure 4C-D. To be able to trust the assay in these conditions, it should be shown how inhibiting replication affects SIRF. Moreover, controls without HU and/or upon CAF-1 depletion should also be presented to characterize how HIRA behaves in normal replication condition and upon CAF-1 removal. This is essential in supporting the interpretation of the authors that HIRA takes over nucleosome assembly in specific situations.

As it is standard for SIRF assays (Schlacher et al, *JCB* 2018, PMID: 29475976), and as we describe in the Methods section, we normalize the protein-EdU signal to EdU incorporation (biotin-biotin signal). This signal normalization ensures that any unspecific effects of siRNA or drug treatments on DNA synthesis (as measured by EdU incorporation) are not confounding the analysis.

To address the reviewer's comment regarding HIRA levels at replication forks in the absence of stress, we now provide extensive analysis of this in the new **Fig. 4c,d** and the new **Supplementary Fig. S4b** in the revised manuscript.

-The choice of depleting ASF1A over ASF1B (the S-phase specific ASF1 variant) is odd, and ASF1B depletions should be included in this study and taken into account in the interpretations.

We apologize for not clarifying this in our original submission. We chose to investigate ASF1A as opposed to ASF1B since ASF1A, but not ASF1B, was previously shown to interact with HIRA-H3.3 complexes (Tagami et al, *Cell* 2004, PMID: 14718166). In the revised manuscript we clarify this issue (page 10-11).

To address the reviewer's comment, in the revised manuscript, we show in the new **Supplementary Fig. S3c,d** that ASF1B depletion by itself does not impact fork protection in either wildtype or BRCA-deficient cells; Moreover, co-depletion of ASF1A and ASF1B mimics the depletion of ASF1A alone (as it causes fork degradation in wildtype cells, and does not

restore fork protection in BRCA-deficient cells). These findings indicate a specific role for ASF1A, and not for ASF1B, in regulating fork protection.

2) The methods (fiber assays, PLA-based readouts, comet assays) are not sufficiently powerful to demonstrate mechanistic relationships. Some examples of overstatements are here, but in my opinion virtually all statements in the abstract are not sufficiently supported by the data.

We respectfully disagree with the reviewer. As also mentioned above, the methods which we are using are state-of-the-art in the field, and have been used by virtually every recent high-impact publication in the field to draw mechanistic conclusions. This is in fact acknowledged by the other two reviewers, who state that our conclusions are supported by the data.

a. page 11, line 4, the author use the wording “replication-coupled DNA damage”, there is no evidence in their data that the effects observed is pertinent to replication-dependent DNA damage. It is only inferred, but not directly tested/shown.

We use the wording “replication-coupled DNA damage” since the drugs we are using (cisplatin, camptothecin, hydroxyurea) have been clearly demonstrated to cause DNA damage in a replication-dependent manner. Moreover, cisplatin and camptothecin have previously been shown to elicit fork degradation in BRCA-deficient cells (Taglialatela et al, *Mol. Cell* 2017, PMID: 29053959; Quinet et al, *Mol. Cell* 2020, PMID: 31676232). We have now added additional data showing that camptothecin treatment causes fork degradation in BRCA-deficient cells, which is rescued upon CHAF1A depletion (new **Supplementary Fig. S2b**), implying a role for CAF-1 loss in preventing camptothecin-associated DNA damage during replication.

b. PLA signal indicates proximity within 40 nm, this corresponds to more than 3 nucleosomes, or 2 replisomes, etc. It basically can not be used to assess “recruitment” as the author argue throughout their manuscript. Experiments like iPOND could be used if the authors want to claim replication-dependencies and recruitment.

Unless one performs iPOND-SILAC experiments which are very tedious and require a specialized mass-spec facility, the iPOND experiments with detection of bound proteins by western blot or mass-spec are in fact less quantitative than SIRF experiments (Schlacher et al, *JCB* 2018, PMID: 29475976). In SIRF experiments, we are able to quantify the signal from individual cells, and derive complete datasets which are analyzed through powerful statistical methods. This is not the case for iPOND experiments, which give a general picture for the whole pool of cells analyzed. The term “recruitment” when referring to SIRF signal has been extensively used in the recent literature, by multiple labs (eg. Taglialatela et al, *Mol. Cell* 2017, PMID: 29053959; Park et al, *Nature Comm* 2019, PMID: 31844045; Nieminuszczy et al, *Mol. Cell* 2019, PMID: 31255466; Kim et al, *Mol. Cell* 2020, PMID: 32966758; Tsai et al, *PLoS Genetics* 2021, PMID: 33826602)

To show that the recruitment and dissociation of proteins measured by SIRF is similar to that observed by iPOND-SILAC, we provide below a Figure for the Reviewer where we analyze the SIRF signal of a number of proteins, under similar conditions as they were previously investigated in iPOND (Sirbu et al, *JBC* 2013, PMID: 24047897; Dugrawala et al, *Mol. Cell* 2015, PMID: 26365379). Our results with SIRF match perfectly the previously-described iPOND results -indicating that this approach is consistently measuring protein recruitment and dissociation at nascent DNA.

Figure for Reviewer 1. SIRF experiment with representative chromatin proteins (PCNA, PolEpsilon, KU, Histone H3) using similar conditions as in previously-published iPOND experiments. Similar patterns of chromatin binding are detected in this SIRF experiment as in the previously-published iPOND experiments.

3) The authors often compare many (i.e. more than 2) conditions in their experiments using statistical tools that are meant to compare only 2 groups (T-test or Mann-Whitney). For example, in figure 2B a T-test is used for the statistical analysis. T-test can only be used to compare a maximum of two groups. As in this figure multiple comparisons of 6 different conditions (so 6 groups) are done, also a test for multiple comparison, like the 2-way Anova, should be performed. Therefore, we recommend that these analyses are redone with the appropriate statistical approaches.

We respectfully disagree with the reviewer. We have used the appropriate statistical analyses, as previously employed in the literature, for all experiments presented. In the example given by the reviewer (Fig. 2b), we are not comparing 6 different conditions (there would be no reason to do that); instead, we are comparing each condition pairwise to siControl (since that is the relevant comparison), using the appropriate statistical test (namely the t-test), and are clearly indicating on the graph what are the pairs being compared.

In any case, we provide the exact values plotted in the Source Data file uploaded with the revised manuscript. This will allow any reader to perform any statistical analyses deemed appropriate.

4) The rationale for the experiments shown in Figure 7 was not clear to me. It is also not clear why these mutants lead to an increase in CldU/IdU ratio shown in Figure 7G (compare siCTRL for the mutants vs the WT PCNA samples). These unexpected variations again argue a limit in the robustness of the fiber assay in deriving the conclusions that the authors are making.

We apologize for not better explaining the rationale for these experiments: In order to demonstrate that PCNA unloading defects represent the root cause for fork degradation in BRCA-deficient cells (by sequestering CAF-1 on ssDNA gaps on the lagging strand), we created PCNA mutants which are unable to accumulate on DNA (as demonstrated in Fig. 7d,e). Since these mutants spontaneously dissociate from DNA, we expected that in these mutants CAF-1 sequestration is ameliorated, allowing CAF-1 recycling to new replication forks and thus restoring fork protection in BRCA-deficient cells. This is exactly what we observed in the fork protection assays in Figs. 7f,g thus validating our model presented in Fig. 7h. In the revised manuscript, we more carefully describe this figure.

We believe that the reviewer missed that in Fig. 7g all samples were treated with the RAD51 inhibitor B02 (as indicated in the labeling scheme presented at the top of the graph), to mimic BRCA deficiency. In fact, in the absence of this treatment, in wildtype cells, the PCNA mutants show no impact on fork degradation compared to wildtype PCNA -this is shown in Fig. 7f. However, in BRCA-deficient cells, the PCNA mutants rescue fork degradation (also shown in Fig. 7f). Since all samples in Fig. 7g were treated with B02, the siControl conditions in this figure reflect the situation in BRCA-deficient cells, where the PCNA mutants rescue the fork degradation. Therefore, this is by no means an “unexpected variation”, but, quite the opposite, it is the expected result and demonstrates that the data obtained from DNA fiber assays is very robust as the PCNA mutants show the same phenotype in the context of BRCA2-knockout (Fig. 7f) as in the context of B02 treatment (Fig. 7g).

Reviewer #2

We are glad that the reviewer found that our study is “*interesting*” and that “*the conclusions are largely supported by data presented*”. We thank the reviewer for their helpful comments, which we have addressed as indicated below.

(Remarks to the Author):

In this manuscript, Thakar et al uncovered a role of CAF-1 mediated nucleosome assembly in BRCA1 mediated fork protection. It is known that previously that depletion of CHAF1A, the gene encoding the large subunit of CAF-1, resulted in degradation of newly synthesized DNA, in wild type cells. Similarly BRCA1 deficiency also led to fork degradation. Interestingly, they found that depletion of CHAF1A in BRCA1 cells suppresses fork degradation. They went on to show the restoration of fork degradation in BRCA1 cells is due to sequestering PCNA and CAF-1 on ssDNA gaps, and depletion of CAF-1 in BRCA1 cells activates the HIRA mediated nucleosome assembly pathway for fork protection. Overall, this is an interesting study and the conclusions are largely supported by data presented. The following concerns should be addressed/clarified before acceptance of this study.

Major concerns:

1) In the figure 1, fork protection was performed in cells treated with HU. In figure 2, the effects on fork protection on genome stability were performed in cells treated with CPT. It is not clear whether these two different drugs will have the same effects on fork degradation in BRCA1 and BRCA2 deficient cells.

In the revised manuscript, we now also include fork protection assays in cells treated with CPT (new **Supplementary Fig. S2b**). Just like in the case of HU treatment, CHAF1A depletion restores fork protection in BRCA1- and BRCA2-deficient cells upon CPT treatment.

2) *Fig. 3A. The effects of Asf1a depletion on fork degradation in wild type and BRCA1 mutant cells need to be revisited. What are the effects of depletion of Asf1b alone and in combination with Asf1a depletion?*

In the revised manuscript, we show in the new **Supplementary Fig. S3c,d** that ASF1B depletion by itself does not impact fork protection in either wildtype or BRCA-deficient cells; Moreover, co-depletion of ASF1A and ASF1B mimics the depletion of ASF1A alone (as it causes fork degradation in wildtype cells, and does not restore fork protection in BRCA-deficient cells). These findings indicate a specific role for ASF1A, and not for ASF1B, in regulating fork protection. This is in fact in line with previously published findings that ASF1A, but not ASF1B, interacts with HIRA and H3.3 (Tagami et al, *Cell* 2004, PMID: 14718166).

3) *What is the effects of Asf1a and HIRA depletion on nascent fork protection in wild type cells upon depletion of CHAF1A?*

In the revised manuscript, we show in the new **Supplementary Fig. S3c** that depletion of ASF1A or HIRA does not further affect fork degradation caused by CHAF1A depletion in BRCA-wildtype cells. Indeed, this result is expected, since ASF1A is upstream of CAF-1 and thus epistatic for fork protection, and HIRA is not involved in fork protection in wildtype cells (but instead is recruited to ssDNA gaps formed in BRCA-deficient cells).

4). *Fig. 5. It is not clear whether the following statement “these findings suggest that the retention of CAF-1 at ssDNA gaps behind the replication fork reduces its availability at ongoing replication forks, causing impaired nucleosome assembly which drives fork degradation in BRCA-deficient cells” is true/accurate for the following reasons. HU, which can slow down replication forks, is used for all experiments to measure fork stability. It is not clear why CAF-1 is needed for ongoing replication forks without movements under HU.*

As mentioned above, in the revised manuscript we show that a similar effect on fork protection can be observed upon CPT treatment, which arrests forks through a different mechanism than HU (new **Supplementary Fig. S2b**). We believe that these findings strengthen our model explained in the statement above. Moreover, even with HU treatment, it is likely that forks are not immediately arrested, but rather slowed. It has been shown that under these conditions, there is an accumulation of ssDNA gaps caused by reactive oxygen species induced by HU treatment (Somyajit et al, *Dev. Cell* 2021, PMID: 33621493). We believe that these gaps are causing CAF-1 retention and subsequent fork degradation upon fork arrest.

5). *It is not clear why both CHAF1A depletion and BRCA1 deficiency are needed for activation of the HIRA-H3.3 pathway.*

We agree with the reviewer that this is an important question. In fact, in the revised manuscript we provide further evidence that this is indeed the case: By analyzing histone H3 binding to nascent DNA by SIRF, we now show that BRCA-deficient cells are defective in H3 deposition on newly-replicated DNA; CHAF1A depletion in these cells restores normal histone H3 levels, and this is dependent on HIRA as co-depletion of HIRA brings H3 levels back down again (new **Fig. 4f**).

As we describe in the manuscript, HIRA loading on nascent DNA requires RPA (Fig. 6), thus we believe that HIRA is recruited to ssDNA gaps, which only occur at an increased frequency in BRCA-deficient cells during replication stress. However, as pointed out by the reviewer, this is not enough for HIRA to be activated; instead, loss of CAF-1 is also needed. We speculate that CAF-1 inhibits the interaction between ASF1A and HIRA, as CAF-1-ASF1A and HIRA-ASF1A interactions are thought to be mutually exclusive (Tang et al, *Nat. Struct. Mol. Biol.* 2006, PMID: 16980972). In the revised manuscript, we present evidence that this is indeed the case, since loss of CHAF1A triggers an increase in the interaction between ASF1A and HIRA as investigated by proximity ligation assays (new **Fig. 4g, Supplementary Fig. S4i**).

6). *What is the reason for the degradation of nascent DNA, but not template DNA in cells with depletion of CHAF1?*

In fact, it cannot be excluded that the template strand is also degraded. However, a limitation of the DNA fiber assay is that we are unable to assay for template strand integrity, since it is not labeled. Only the nascent strand is labeled by CldU/IdU in the DNA fiber combing assay, and we can detect its degradation, therefore we refer to it as “nascent DNA degradation”.

Minor concerns:

Fig. S6A and S6B, the effects of two siRNA against PRIMPOL on ssDNA gaps are very different, which is in contrast to similar effects reported in Fig. 6A-B using another assay.

In the Supplementary Fig. S6b, both PRIMPOL siRNAs cause a reduction in CldU tract length in the absence of S1 treatment, albeit we agreed that the reduction is more severe with siRNA#1 compared to siRNA#2. While we do not know the reason for this small difference, both siRNAs in fact show similar results in respect to ssDNA gaps accumulation, since for both of them S1 treatment does not further reduce the CldU tract length (unlike the siControl sample, where S1 treatment causes a significant reduction in CldU tract length).

Reviewer #3

We are glad that the reviewer found that our studies are “*well-presented*” and “*reveal a novel role for histone deposition*”, and thus our work “*is of high interest to the DNA damage, replication and repair communities*”. We thank the reviewer for their helpful comments, which we have addressed as indicated below.

(Remarks to the Author):

In this manuscript, Thakar et al. report that the sequestration of the histone chaperone CAF-1 at ssDNA gaps causes nascent DNA degradation in BRCA1/2-deficient cells. In addition, they show that loss of CAF-1 suppresses fork degradation in BRCA1/2-deficient cells as a result of compensatory activation of HIRA-dependent histone deposition. These well-presented studies reveal a novel role for histone deposition in controlling the stability of stalled replication forks in BRCA1/2-deficient cells. As such, this work is of high interest to the DNA damage, replication and repair communities. We recommend the authors to address the points below to further strengthen the findings of their manuscript.

1) The authors show in Figure 2 that loss of CAF-1 reduces CPT-induced genome instability and cisplatin-mediated loss of viability in BRCA1/2-deficient cells. It would be important to also determine whether CAF-1 loss reduces cisplatin-induced genome instability in BRCA1/2-deficient cells.

In the revised manuscript, we now show that CAF-1 loss also reduces cisplatin-induced genome instability in BRCA-deficient cells (new Fig. 2e).

2) The authors show that LIG1 deficiency induces Okazaki fragment maturation defects in BRCA1/2-deficient cells (Figure 6D). Does LIG1 loss enhance CAF-1 retention at nascent DNA and fork degradation in BRCA1/2-deficient cells? Are similar phenotypes also observed upon LIG3 loss, which was recently shown to cause gap accumulation in BRCA1-deficient cells (Dias et al, 2021)?

In the revised manuscript we show that LIG1 depletion enhances CAF-1 retention at nascent DNA in wildtype cells, but does not further increase CAF-1 retention in BRCA-deficient cells (new Supplementary Fig. S7c). These findings are in line with our model: In wildtype cells, LIG1 loss causes ssDNA nicks which impair OF maturation and PCNA unloading, thus sequestering CAF-1. In BRCA-deficient cells, OF maturation is already impaired because of ssDNA gap accumulation, thus loss of LIG1 does not have any additional effect. In accordance with this, we moreover also show that LIG1 depletion causes fork degradation in wildtype cells, and does not further enhance the fork degradation observed in BRCA-deficient cells (new Supplementary Fig. S7d).

Regarding LIG3, in the revised manuscript we show that loss of LIG3 has a similar impact in both wildtype and BRCA-deficient cells (a mild 3-4 fold increase in CAF-1 retention), and does not affect fork protection in either wildtype or BRCA-deficient cells (new Supplementary Fig. S7c-e), suggesting that LIG3 is not directly involved in CAF-1-mediated fork protection.

3) Given that PRIMPOL loss partially reduces gap formation in BRCA1/2-deficient cells (Figure 6B), it would be informative to determine whether it may have an impact on CAF-1 retention at nascent DNA in BRCA1/2-deficient cells.

In the revised manuscript, we now show that loss of PRIMPOL does not affect CAF-1 retention on nascent DNA in BRCA-deficient cells; in contrast, Pol α inhibition suppresses it (new Supplementary Fig. S6g). These findings are in line with our model that CAF-1 is retained on lagging strand gaps, formed upon repriming by Pol α but independent of PRIMPOL.

4) *Since ATAD5 loss enhances PCNA levels on chromatin (Figure 7D), it would be useful to show whether it increases CAF-1 retention at nascent DNA and fork degradation in BRCA1/2-deficient cells. Can ATAD5 overexpression reduce nascent DNA degradation in BRCA1/2-deficient cells?*

In the revised manuscript, we now show that, similar to **LIG1**, ATAD5 depletion increases CAF-1 retention on nascent DNA in wildtype cells but not in BRCA-deficient cells, and causes fork degradation in wildtype cells but does not affect fork degradation in BRCA-deficient cells (new **Supplementary Fig. S7c,d**). These findings are in line with our model that ATAD5-mediated PCNA unloading is needed for CAF-1 recycling to ongoing replication forks in wildtype cells; in BRCA-deficient cells, PCNA unloading is already compromised because of its retention on ssDNA gaps which preclude OF maturation, thus ATAD5 loss does not further impact it.

Regarding ATAD5 overexpression, despite our efforts we were unfortunately not able to obtain overexpressing cell lines during the course of this revision, due to technical challenges (since ATAD5 is a very large protein, of 1,844aa and 207kDa). In any case, it is not clear if ATAD5 overexpression by itself would be functional and cause PCNA unloading, since ATAD5 forms a PCNA-unloading complex with RFC2-5 subunits of the RFC complex, making this kind of overexpression experiments very challenging.

5) *Page 13. “tend to dependent” should be “tend to be dependent”.*

We thank the reviewer for pointing out this error, and corrected it in the revised manuscript.

REVIEWERS' COMMENTS

Reviewer #1 (Remarks to the Author):

The revised article by Thakar et al has addressed some of the points that were made by the reviewers, containing clarifications and additional data. The data remains highly correlative and the authors have opted not to include independent approaches to test their hypotheses (i.e. iPOND). In my view this is a great limit of this work, with the conclusions remaining greatly overstated. I appreciate that the fiber and PLA assays presented can however be useful for the community.

I want to note that it is the authors', not the readers', responsibility to publish the correct statistical analyses of their data, as this may influence the conclusions. I remain convinced from any biostatistics textbook that ANOVA is a better test for significance in pairwise-comparison of groups that are part of multi-group measurements, as this includes variance analysis of all groups.

Reviewer #2 (Remarks to the Author):

The revised manuscript largely addressed my concerns, and I support its acceptance.

Reviewer #3 (Remarks to the Author):

The authors have satisfactorily addressed the concerns of this reviewer. We therefore recommend the manuscript for publication..

Response to referees

We are happy that the reviewers found that our revised manuscript satisfactorily addressed their comments, and recommended the manuscript for publication. Below, please find our responses to the reviewer's remaining comments (**our responses in red font**).

Reviewer #1

The revised article by Thakar et al has addressed some of the points that were made by the reviewers, containing clarifications and additional data. The data remains highly correlative and the authors have opted not to include independent approaches to test their hypotheses (i.e. iPOND). In my view this is a great limit of this work, with the conclusions remaining greatly overstated. I appreciate that the fiber and PLA assays presented can however be useful for the community.

We thank the reviewer for their comments, and are glad to learn that the reviewer found that our revised manuscript was improved. We now include in the Discussion section an acknowledgement of the fact that our manuscript does not contain iPOND approaches to independently confirm our proximity ligation-based assays results (page 24).

I want to note that it is the authors', not the readers', responsibility to publish the correct statistical analyses of their data, as this may influence the conclusions. I remain convinced from any biostatistics textbook that ANOVA is a better test for significance in pairwise-comparison of groups that are part of multi-group measurements, as this includes variance analysis of all groups.

We now include, in the Source Data file, the results of additional statistical analyses, which align well to our original analyses. For experiments analyzed with the t-test (such as comet assays), we now also provide statistical analyses using one-way ANOVA with Holm-Sidak's multiple comparison test. Moreover, for experiments analyzed with the Mann-Whitney test (such as DNA fiber and proximity ligation assays), we now also provide Kruskal-Wallis with Dunn's multiple comparison analyses. These additional statistical analyses are provided in the Source Data file for Figs. 1a, 1b, 1c, 1d, 1f, 1g, 1h, 1i, 2b, 2d, 2e, 3a, 3b, 3c, 4d, 4e, 4f, 4g, 5a, 5b, 5c, 5e, 5f, 5g, 6b, 6d, 6f, 6h, 7b, 7c, and Supplementary Figs. S2b, S3c, S4c, S4e, S4g, S5a, S6b, S7a, S7d.

Reviewer #2

The revised manuscript largely addressed my concerns, and I support its acceptance.

We are happy to learn that the reviewer found that our revised manuscript largely addressed their concerns, and recommended its acceptance.

Reviewer #3

The authors have satisfactorily addressed the concerns of this reviewer. We therefore

recommend the manuscript for publication.

We are happy to learn that the reviewer found that our revised manuscript addressed their concerns, and recommended its acceptance.